# A single-cell transcriptional landscape of immune cells shows disease-specific changes of T cell and macrophage populations in human achalasia

Zu-Qiang Liu[1,2,6], Hao Dai [3,6], Lu Yao[1,2,6], Wei-Feng Chen[1,2,6], Yun Wang[1,2], Li-Yun Ma[1,2], Xiao-Qing Li[1,2], Sheng-Li Lin[1,2], Meng-Jiang He[1,2], Ping-Ting Gao[1,2], Xin-Yang Liu[1,2], Jia-Xin Xu[1,2], Xiao-Yue Xu[1,2], Ke-Hao Wang[1,2], Li Wang[1,2], Luonan Chen [3,4,5] ✉, Ping-Hong Zhou [1,2] ✉ & Quan-Lin Li [1,2] ✉

Achalasia is a rare motility disorder of the esophagus caused by the gradual degeneration of myenteric neurons. Immune-mediated ganglionitis has been proposed to underlie the loss of myenteric neurons. Here, we measure the immune cell transcriptional profile of paired lower esophageal sphincter (LES) tissue and blood samples in achalasia and controls using single-cell RNA sequencing (scRNA-seq). In achalasia, we identify a pattern of expanded immune cells and a specific transcriptional phenotype, especially in LES tissue. We show C1QC[+] macrophages and tissue-resident memory T cells (T$_{RM}$), especially ZNF683[+] CD8[+] T$_{RM}$ and XCL1[+] CD4[+] T$_{RM}$, are significantly expanded and localized surrounding the myenteric plexus in the LES tissue of achalasia. C1QC[+] macrophages are transcriptionally similar to microglia of the central nervous system and have a neurodegenerative dysfunctional phenotype in achalasia. T$_{RM}$ also expresses transcripts of dysregulated immune responses in achalasia. Moreover, inflammation increases with disease progression since immune cells are more activated in type I compared with type II achalasia. Thus, we profile the immune cell transcriptional landscape and identify C1QC[+] macrophages and T$_{RM}$ as disease-associated immune cell subsets in achalasia.

Achalasia is a rare motility disorder of the esophagus with symptoms of dysphagia, regurgitation, chest pain, and weight loss[1]. The pathophysiology of achalasia is aberrant esophageal peristalsis and impaired relaxation of the lower esophageal sphincter (LES) caused by the gradual degeneration of the myenteric neuron. Although its etiology is unknown, immune-mediated ganglionitis triggered by environmental factors and genetic susceptibilities may underlie the loss of myenteric neurons in achalasia[1,2]. Previous studies demonstrated various inflammatory cells infiltrated in LES, such as T cells, eosinophils, and mast cells[2-4]. Whole-exome and RNA sequencing of achalasia patients

[1]Endoscopy Center and Endoscopy Research Institute, Zhongshan Hospital, Fudan University, Shanghai, China. [2]Shanghai Collaborative Innovation Center of Endoscopy, Shanghai, China. [3]Key Laboratory of Systems Biology, Shanghai Institute of Biochemistry and Cell Biology, Center for Excellence in Molecular Cell Science, Chinese Academy of Sciences, Shanghai, China. [4]Key Laboratory of Systems Health Science of Zhejiang Province, Hangzhou Institute for Advanced Study, University of Chinese Academy of Sciences, Chinese Academy of Sciences, Hangzhou, China. [5]School of Life Science and Technology, ShanghaiTech University, Shanghai, China. [6]These authors contributed equally: Zu-Qiang Liu, Hao Dai, Lu Yao, Wei-Feng Chen. ✉e-mail: lnchen@sibs.ac.cn; zhou.pinghong@zs-hospital.sh.cn; liquanlin321@126.com

identified achalasia-associated loci enriched for immunological and neurological processes[5,6], suggesting that the immune response may be a critical factor in the disease. However, most studies have focused on the proportion of immune cells and inflammatory mediators without exploring the underlying mechanism. Without cell-type-specific gene expression profiles, it is difficult to define the cell types responsible for achalasia-related changes.

Single-cell RNA sequencing (scRNA-seq), which provides an unbiased approach for characterizing cell diversity and heterogeneous phenotypes at high resolution[7,8], has been used in patients with neurodegenerative disorders or neuroinflammatory conditions, such as Alzheimer's disease (AD)[9], multiple sclerosis (MS)[10], and Parkinson's disease (PD)[11]. Since the loss of myenteric neurons and ganglionitis was the main pathophysiology in achalasia, we use scRNA-seq to map the immune cell transcriptional landscape of peripheral blood and paired LES tissues from patients with achalasia and controls. In this study, we find a specific composition and transcriptional phenotype of C1QC+ macrophages and tissue-resident memory T cells (T$_{RM}$) in achalasia, which might be involved in the inflammatory process and pathophysiology of achalasia.

## Results

### Single-cell transcriptomics of immune cells in tissue and blood of achalasia patients and controls

To generate a deep transcriptional landscape of immune cells in achalasia, we profiled scRNA-seq and coupled T cell receptor (TCR) and B cell receptor (BCR) sequencing. Nine achalasia patients and four controls with paired tissue and blood samples, and two achalasia patients with only peripheral blood samples were included in this study (Fig. 1a and Supplementary Data 1). Immune cells in tissue were dissociated from smooth muscle in LES, which is the target tissue of achalasia. LES smooth muscle specimens of achalasia patients were obtained during the peroral endoscopic myotomy procedure. Patients with benign leiomyomas originating from LES served as controls. After the leiomyomas were removed by submucosal tunneling endoscopic resection, the control tissue specimens were taken from the surrounding normal tissue without tumor invasion. After quality control and filtering, raw data were obtained for 137,346 cells for further analysis, including 71,450 peripheral blood mononuclear cells (PBMC) and 65,896 tissue single-cell transcriptomes. These cells had a median of 1500 genes per cell.

For immune cell types, we integrated the raw data using fast mutual nearest neighbors (MNN) to correct batch effects and used an unsupervised graph-based clustering algorithm implemented in Seurat 4 to distinguish the populations. After first-round clustering, we identified the major cell types by their canonical markers, including immune cells (lymphocytes and myeloid cells) and non-immune cells (endothelial cells, fibroblasts, smooth muscle cells, hematopoietic stem and progenitor cells [HSPCs], and platelets) (Fig. 1b–d and Supplementary Data 2).

Analysis of the proportions of immune cells in tissue compared with blood showed different tissue preferences (Fig. 1e, f). Macrophages, CLEC9A+ conventional dendritic cells (cDCs), CD1C+ cDCs, and CD8+ T cells were highly enriched in LES tissue, and B cells, FCGR3A+ monocytes, NK cells, NKT cells, and CD4+ T cells were enriched in blood. The CD14+ monocytes and γδ T cells were comparable between tissue and blood (Fig. 1e, f). The tissue-resident cells, including macrophages, mast cells, endothelial cells, fibroblasts, and smooth muscle cells, were almost absent in peripheral blood (Fig. 1b, e), demonstrating the accuracy of the data analysis. Although without a statistical difference, the proportions of CD4+ T cells and CD8+ T cells in LES tissue were slightly increased in achalasia compared with those in controls, while the proportion of each cell type in peripheral blood was quite similar between achalasia and controls. (Fig. 1e and Supplementary Data 1). We, therefore, mapped the landscape of the major immune cells from each individual by scRNA-seq.

To conduct a high-dimensional validation of the scRNA-seq landscape, we measured the expression of 42 surface and intracellular markers on immune cells in the LES tissue from achalasia and controls using time-of-flight mass cytometry (CyTOF). We identified the major myeloid cells and lymphocytes in the LES tissue by t-SNE analysis of canonical markers and analyzed the cell proportions of immune cells of achalasia and controls (Supplementary Fig. 1a–c and Supplementary Data 3). The proportions of CD8+ T cells in the LES tissue were slightly increased in achalasia compared with those in controls.

### Single-cell transcriptomics of subclustered myeloid cells

By unsupervised subclustering of myeloid cells, we identified 11 clusters, including seven clusters of monocytes/macrophages and three clusters of dendritic cells (DCs), and one cluster of mast cells (Fig. 2a, b). Based on canonical markers, the monocytes/macrophages were identified as Macro1 (C1QC+), Macro2 (CXCL3+), Macro3 (FOS$^{high}$), Macro4 (CCL20+), Mono1 (CD14+ S100A8 high, classical monocytes), Mono2 (CD14+ ITGA4 high, intermediate monocytes), and Mono3 (FCGR3A [CD16]+, non-classical monocytes) (Fig. 2c, d, Supplementary Fig. 2c, and Supplementary Data 2 and 4). Based on previously reported signatures[12], we did not find definitely enriched dichotomy of classically activated (M1) or alternatively activated (M2) phenotypes for the macrophage clusters, indicating that macrophages had more complicated activating states than M1/2 polarization models (Supplementary Fig. 2a). The pseudotemporal trajectory of monocytes/macrophages showed three main directional flows that originated from classical monocytes to intermediate/non-classical monocytes, C1QC+ or CXCL3+ macrophages, and FOS$^{high}$ or CCL20+ macrophages, which indicated the differentiation hierarchies for myeloid cells. This result was consistent based on different pseudo-trajectory inference algorithms (Supplementary Fig. 2b). Previously, myeloid-derived suppressor cells (MDSCs) were reported with high expressions of S100A family genes and relatively low expressions of HLAs[13,14]. We found all the monocytes and macrophages, except C1QC+ and CXCL3+ macrophages, highly expressed S100A family genes of S100A8, S100A9, FCN1 and VCAN, and relatively low expressed HLAs (Fig. 2d and Supplementary Fig. 2d), which indicated all the monocytes and macrophages, except C1QC+ and CXCL3+ macrophages, were MDSC-like monocytes/macrophages. Three clusters of DCs, including plasmacytoid DCs (pDCs) and 2 cDCs (cDC1 and cDC2), were characterized by high HLA-DRs and low CD14 expression. Among them, pDCs were identified by the expression of CLEC4C and LILRA4; cDC1 were identified by the expression of CLEC9A, XCR1 and BATF3; cDC2 were identified by the expression of CD1C, FCER1A and CD1A. Mast cells were distinguished by the expression of TPSAB1, KIT, MS4A2 and CPA3 (Supplementary Fig. 2c).

### C1QC+ macrophages were significantly expanded and associated with multiple neurodegenerative and behavioral disorder pathways in achalasia

Next, achalasia-associated compositional changes of myeloid cells were explored. Myeloid cells in blood were enriched to comparable levels in achalasia and controls (Fig. 2b), but the cell compositions in tissue were clearly different, with a significant expansion of C1QC+ macrophages in achalasia compared with controls (Fig. 2b and Supplementary Data 4). Besides, multicolor IHC also confirmed that C1QC+ macrophages were expanded and located surrounding or possibly infiltrating the myenteric plexus (shown by PGP9.5) of LES in achalasia (Fig. 2e), suggesting a potential correlation with the enteric nervous system (ENS). Further CyTOF analysis showed the C1QC+ macrophages had a tendency of increased proportion in the LES of achalasia compared with controls (Supplementary Fig. 1d).

We also analyzed disease-associated transcriptional changes. Analysis of differentially expressed genes (DEGs) between achalasia

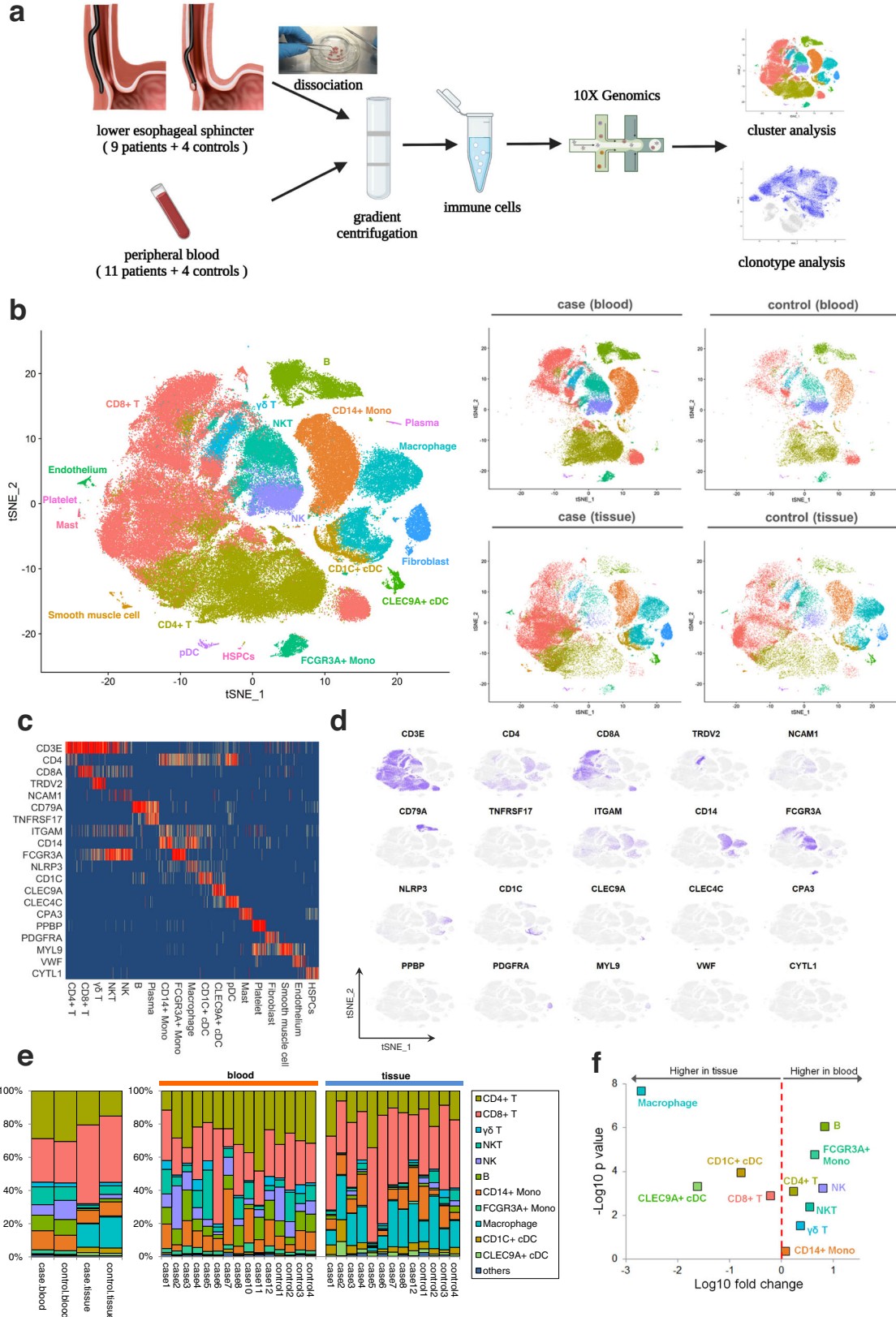

and controls showed the co-expression of many *human leukocyte antigens* (*HLAs*) and upregulations of *small ribosomal subunit* (*RPS*) and *large ribosomal subunit* (*RPL*) genes in multiple myeloid cells (Fig. 2f and Supplementary Data 4), similar to activated microglia during nerve injury[10,15]. This pattern of expression indicated increased ribosomal biogenesis and protein synthesis with the activation of the

myeloid cells[10,15]. Kyoto Encyclopedia of Genes and Genomes (KEGG) analysis showed the expression of pathways of antigen processing and presentation and phagosome in almost all the myeloid cells (Fig. 2g). However, multiple neurodegenerative and behavioral disorders pathways were associated with C1QC+ macrophages and cDC2, including pathways of neurodegeneration-multiple diseases, AD, PD, and others.

**Fig. 1 | The transcriptional landscape of single immune cells in tissue and blood from achalasia and controls. a** A schematic representation of the experimental design and analysis with tissue sources, sorting strategy, and scRNA-seq results. Immune cells in tissue were dissociated from smooth muscle in the LES. The LES specimens were collected from patients with achalasia and benign esophageal leiomyoma (as controls) undergoing endoscopic procedure. After the leiomyomas were removed, the control tissue specimens were taken from the surrounding normal tissue 5 mm away from the leiomyoma without tumor invasion. **b** t-SNE plots of the major immune and non-immune cells found in LES tissue and blood. **c, d** Heatmap (**c**) and t-SNE plots (**d**) of the relative expressions of canonical marker genes for major immune and non-immune cells. **e** Major cluster proportions for LES tissue and blood between achalasia and controls, colored by cell type. **f** Major cluster proportions in LES tissue compared with blood. Source data are provided as a Source Data file.

Multiple metabolic pathways were enriched in the C1QC[+] macrophages, especially oxidative phosphorylation, which were also highly expressed in cDC2. Moreover, KEGG enrichment analysis of the differentially expressed proteins detected by mass spectrometry of bulk LES tissue confirmed the KEGG pathways of C1QC[+] macrophages (Fig. 2h and Supplementary Data 5). Collectively, we found C1QC[+] macrophages were significantly expanded and associated with multiple neurodegenerative and behavioral disorder pathways in achalasia.

## C1QC[+] macrophages were transcriptionally similar to neuronal-associated gut-resident macrophages

Among the macrophages, gut-resident macrophages are resident within the gastrointestinal tract and have various functions based on the niche they occupy[16]. Recent advances have revealed transcriptional signatures of gut-resident macrophages in human and mouse[17,18] and found CD4 and TIM4 are canonical markers for gut-resident macrophages and crucial for the differentiation and development of macrophages[18]. We found C1QC[+] macrophages expressed transcriptional gene set similar to those of previously reported gut-resident macrophages in human and mouse[17,18] (Supplementary Fig. 3a, b). Besides, C1QC[+] macrophages transcriptionally highly expressed *CD4* and *TIM4* and were validated by multicolor IHC (Supplementary Fig. 3c, d). Both indicated C1QC[+] macrophages might be gut-resident macrophages.

Among the gut-resident macrophages, one cluster of gut-resident macrophages named neuronal-associated gut-resident macrophages, close to enteric neurons, could support enteric neurons and control essential intestinal functions of motility and secretion[19,20]. We found C1QC[+] macrophages expressed transcriptional gene set similar to those of neuronal-associated gut-resident macrophages[19,20] (Supplementary Fig. 4a, b), indicating the potential impact on C1QC[+] macrophages on enteric neurons.

## C1QC[+] macrophages were transcriptionally similar to microglia and exhibited a neurodegenerative dysfunctional phenotype in achalasia

Since neuronal-associated gut-resident macrophages were reported to be similar to cerebral microglia[19], we next explored the similarity of C1QC[+] macrophages and microglia. We found that C1QC[+] macrophages highly expressed previously identified human microglial gene sets[21,22], including *C1QA, C1QB, C1QC, TREM2, TMEM119, P2RY12,* and *GPR34* (Fig. 3a). Metascape analysis of marker genes of C1QC[+] macrophages showed enrichment in the microglial pathogen phagocytosis pathway (Fig. 3b). Multicolor IHC and t-SNE analysis of CyTOF confirmed this C1QC[+] macrophage cluster highly expressed C1QC, TMEM119, P2RY12, and TREM2 (Fig. 3c and Supplementary Fig. 1d), which are canonical markers of microglia. Microglia are the resident macrophages in the brain and are involved in the homeostasis, plasticity, immunity, and repair of the CNS[23]. Given the transcriptional similarity between C1QC[+] macrophages and microglia and the wide distribution of ENS in the gastrointestinal tract, we speculated that C1QC[+] macrophages might have an impact on the ENS.

Metascape analysis of DEGs showed that C1QC[+] macrophages had increased expression of transcripts of TYROBY causal network in microglia, synapse pruning, phagosome, lysosome, microglial pathogen phagocytosis pathway, and microglial cell activation in achalasia (Fig. 3d). These pathways highlighted C1QC[+] macrophages in achalasia exhibited an activated state similar with microglia. C1QC[+] macrophages also showed high upregulation of *CALR* in achalasia (Fig. 3d), a C1Q receptor gene that could contribute to microglial activation[24–26]. The activated microglial complement-mediated synaptic pruning and loss is highly upregulated in many neurodegenerative disorders[26–28], suggesting that a similar mechanism of complement-mediated synapse elimination might drive achalasia progression.

Dysregulation of microglia contributes to the pathogenesis of several neurodegenerative diseases, including AD, PD, and MS[29–31]. Given the similarities of C1QC[+] macrophages and microglia, we compared transcripts of C1QC[+] macrophages in achalasia with those of disease-associated microglia from other neurodegenerative diseases[9,10,21,29–32]. Gene set variation analysis (GSVA) showed that the signatures of DEGs for C1QC[+] macrophages were similar to those of neurodegenerative dysfunctional microglia (Fig. 3e and Supplementary Data 5). Many disease-associated genes in microglia were differentially expressed in C1QC[+] macrophages in achalasia, including 129 high and 38 low DEGs (Fig. 3f and Supplementary Data 5). Of these, 48 high and 2 low DEGs overlapped with previously reported both human and mouse dysregulated gene sets, including *APOE, CD68, C1QA, C1QC, TREM2, FN1, TYROBP, GRN,* and *CTSB* (Fig. 3g, h). Multiple M0-homeostatic microglial transcripts were repressed in C1QC[+] macrophages, including *CST3, NFKB1, JUN,* and *EGR1*, while the key transcripts of dysfunctional microglia in neurodegenerative diseases, including *APOE, TREM2,* and *LGALS3*, were strongly upregulated[32] (Fig. 3d). Among these, *EGR1* was the master transcription factor in M0-homeostatic microglia, and APOE-TREM2 was the master pathway in controlling the switch from a homeostatic to a neurodegenerative dysfunctional state of microglia during phagocytosis of apoptotic neurons. This suggested the C1QC[+] macrophages exhibited a neurodegenerative dysfunctional phenotype in achalasia, possibly mediating the loss of immune cell homeostasis and neuronal damage.

## Single-cell transcriptomics of subclustered lymphocytes

Unsupervised subclustering of lymphocytes yielded 26 subclusters, including 18 clusters of T cells, three clusters of B cells, one cluster of plasma cells, and four clusters of NK cells (Fig. 4a, b). Based on canonical markers, the CD4[+] and CD8[+] T cell clusters were further categorized into 6 main cell types, including naive T cells (CCR7), central memory T cells (CD27, CD44, CCR7[+]), effector memory T cells (CD27, CD44, CCR7[-]), T regulatory cells (Treg cells) (FOXP3), tissue-resident memory T cells (T$_{RM}$) (CD69), and effector T cells (GNLY) (Fig. 4c, d, and Supplementary Data 2 and 6). The residency marker *CD69*, which restricts lymphocyte tissue egress via *S1P1* inhibition, was also detected in tissue-resident NK and B cells (CD69[+] NK and CD69[+] B cells). We determined the core signatures (29 genes) shared by T$_{RM}$ and tissue-resident NK and B cells, including chemokines (*CXCR4* and *CCL4*), transcription factors (e.g., *JUN, FOS*), heat shock proteins (e.g., *HSP90AA1, HSPA1A*), and zinc fingers (*ZFAND2A* and *ZFP36*) (Fig. 4e). The pseudotemporal trajectory of T cells showed a strong directional flow from the naive to the memory and finally to the effector state (Fig. 4f), as seen previously[33].

## T$_{RM}$ were increasingly infiltrated and surrounding the myenteric plexus in achalasia

Next, achalasia-associated compositional changes in lymphocytes were explored. We found the proportions of circulating lymphocytes were similar between achalasia and controls except for the increased

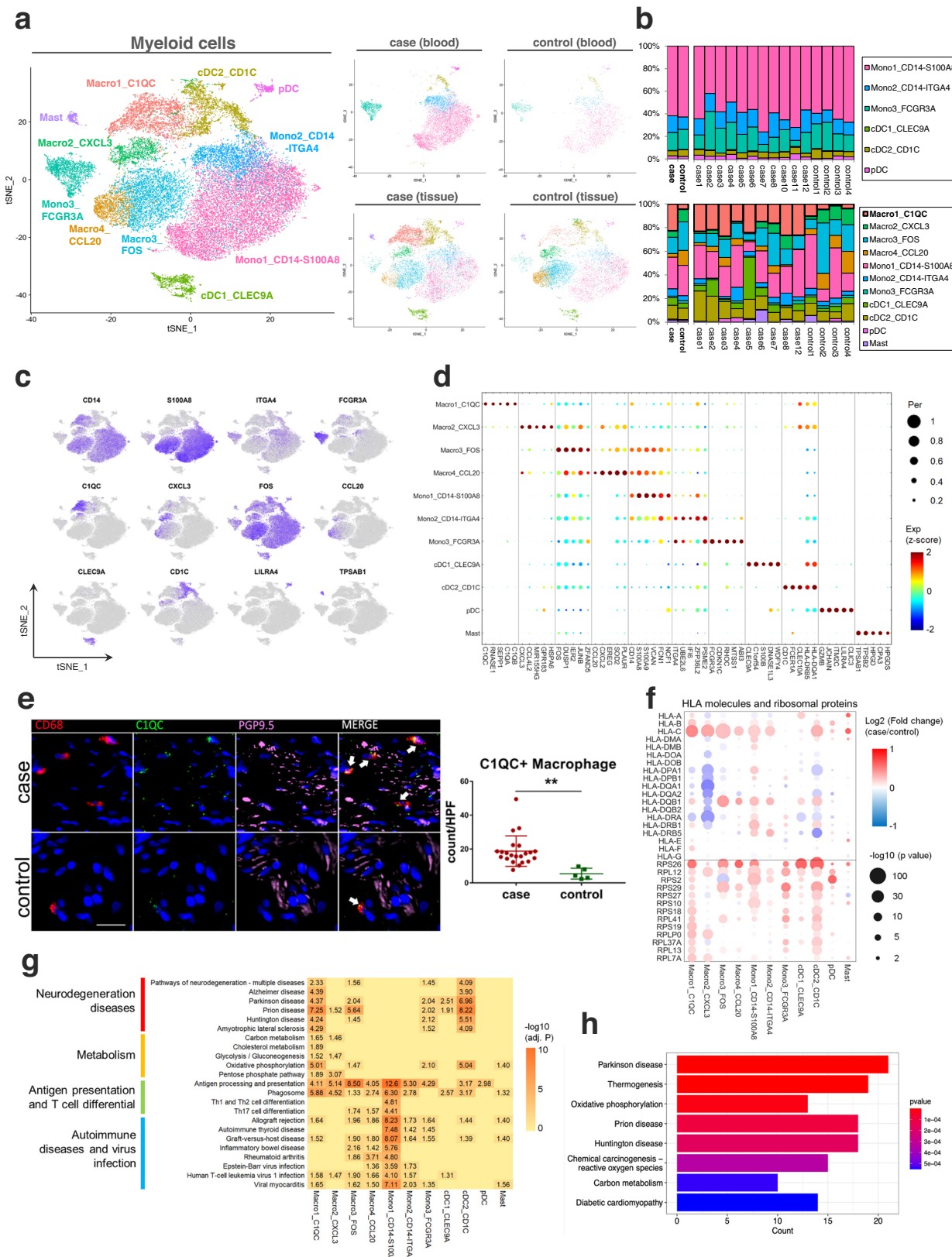

populations of GNLY⁺ CD8⁺ T cells (Tc8), CD27⁺ B cells (B2), and plasma in achalasia. In contrast, proportions of lymphocytes in the LES tissue were clearly different, with a more expansion of $T_{RM}$ in achalasia (Fig. 4d). Among the $T_{RM}$ subclusters, XCL1⁺ CD4⁺ $T_{RM}$ (Th5) and ZNF683⁺ CD8⁺ $T_{RM}$ (Tc5), expanded remarkably in achalasia compared with controls. Multicolor IHC also confirmed the $T_{RM}$ were increasingly

infiltrated and mainly restricted to the area surrounding the sparse myenteric plexus (shown by PGP9.5) in the LES tissue in achalasia compared with those in controls (Fig. 4g, h). Further CyTOF analysis showed an increased tendency of $T_{RM}$ in the LES of achalasia compared with controls (Supplementary Fig. 1d). Since the LES is the target tissue of achalasia, $T_{RM}$ might be involved in the pathophysiology of achalasia.

**Fig. 2 | Single-cell transcriptomics of subclustered myeloid cells. a** t-SNE plots of the 11 myeloid clusters and detailed lineages in different tissue contexts. **b** Myeloid cluster proportions for LES tissue and blood between achalasia and controls, colored by cell types. **c, d** t-SNE plots and dot plot of canonical marker genes for 11 myeloid clusters. **e** Multicolor IHC and quantifications of C1QC⁺ macrophages in the LES tissue between achalasia ($n = 22$) and controls ($n = 5$). The C1QC⁺ macrophages were increasingly infiltrated and surrounding the myenteric plexus (shown by PGP9.5) in achalasia. Scale bar, 20 μm. Data are represented as mean ± standard deviation (SD). Statistics: two-tailed unpaired $t$-test. **P < 0.01. Arrow, C1QC⁺ macrophages. **f** Dot plot showing expression patterns of selected DEGs for 11 myeloid clusters between achalasia and controls. Statistics: two-sided Wilcoxon rank-sum test. **g** KEGG analysis based on DEGs for 11 myeloid cells. Statistics: the adjusted $P$-value was calculated by Fisher's exact test after Benjamini–Hochberg correction. **h** KEGG analysis of differentially expressed proteins between achalasia and controls detected by mass spectrometry of bulk LES tissue. Statistics: Fisher's exact test. Source data are provided as a Source Data file.

## Transcriptome of T_RM

We then aimed to further characterize the T_RM. This study shows that the T_RM are resident in the muscularis propria of the alimentary tract; previously, T_RM were defined as within the epithelial layer and lamina propria[34]. We confirmed that published signatures of T_RM in human and mouse were also highly expressed in the T_RM in our study (Supplementary Fig. 5a)[35–37]. By comparison with other T cells, we identified merged marker genes of T_RM, which were enriched in the pathways of hemostasis and cellular response to stress based on Metascape analysis (Supplementary Fig. 5b and Supplementary Data 6). We found different T_RM subclusters exhibited similar transcriptional patterns compared with other lymphocyte clusters, while different T_RM subclusters also had their own marker genes (Supplementary Fig. 5c, d). The TYMS⁺ CD8⁺ T_RM (Tc7) displayed mitotic features with high expression of genes associated with proliferation, as described previously[38] (Supplementary Fig. 5c). Metascape analysis of marker genes of each T_RM subcluster showed that ZNF683⁺ CD8⁺ T_RM and XCL1⁺ CD4⁺ T_RM, the most common types in achalasia, were enriched mainly in adaptive immune and lymphocyte activation (Supplementary Fig. 5e).

Analysis of the pseudotemporal transcriptional trajectories of CD4⁺ and CD8⁺ T cells showed that most T_RM subclusters originated from effector memory T cells, while TYMS⁺ CD8⁺ T_RM branched out in pseudotime, indicating the specifically proliferated state (Supplementary Fig. 5f, g). Heat maps also showed continuous phenotypic variation along pseudotime and the cell type-specific changes (Supplementary Fig. 5g). The naive and central memory T cell markers, such as *CCR7* and *SELL* were expressed early, while expression of activation markers such as *GNLY*, increased along pseudotime. A series of genes were highly expressed in T_RM, including *XCL1, FOS, FOSB, TSC22D3, JUN, NR4A2*, suggesting their potential functions for T_RM.

## T_RM presented dysregulated immune responses in achalasia

Next, we explored the achalasia-associated transcriptional changes in lymphocytes, especially T_RM. Analysis of differentially expressed transcripts of lymphocytes in achalasia showed that T_RM, GZMK⁺ CD8⁺ T cells (Tc2), and GNLY⁺ CD8⁺ T cells had more DEGs than other lymphocytes (Fig. 5a). We found the T_RM subclusters exhibited quite similar DEGs (Fig. 5b). Among these DEGs, *ZNF683*, important for the differentiation and maintenance of T_RM[39], and *RBPJ*, central to Notch signaling and maintenance of T_RM[40], were highly expressed in T_RM (Fig. 5c), which might explain the high infiltration of T_RM in achalasia. Similar to myeloid cells, multiple ribosome-related genes and *HLAs* were highly expressed in almost all the T_RM in achalasia (Fig. 5d), indicating a proliferated and activated state[41]. Besides, many co-inhibitory molecules, including *PDCD1, HAVCR2, LAG3, ICOS, PRDM1*, and *MAF*, were markedly downregulated in T_RM (Fig. 5d), supporting an activated cell state in achalasia.

GO and KEGG analysis showed T_RM, GZMK⁺ CD8⁺ T cells, and GNLY⁺ CD8⁺ T cells were mutually enriched in several functional categories, including T cell activation, response to interferon-gamma, response to tumor necrosis factor (GO analysis), and Th1 and Th2 cell differentiation, cell adhesion molecules, multiple immune diseases (KEGG analysis) (Fig. 5e). The highly enriched pathways of phagosome and antigen processing and presentation might be the result of the high expression of *HLAs* (Fig. 5d, e), which indicated specific T cell activation[41]. Metascape analysis also confirmed the results of GO and

KEGG analysis (Fig. 5f). Among T_RM subclusters, ZNF683⁺ CD8⁺ T_RM and XCL1⁺ CD4⁺ T_RM were mainly enriched in cell activation, TCR signaling, cytokine signaling in the immune system, and positive regulation of the immune response. However, several cytotoxic enzymes and inflammatory cytokines, including *GZMA, GZMB, GZMK, GNLY*, and *TNF* were downregulated in T_RM in achalasia (Fig. 5d), consistent with a previous study of T_RM in MS[42]. These results indicated that T_RM presented dysregulated immune responses, with some pro-inflammatory pathways upregulated while some others downregulated.

## Clonal expansions but a decreased diversity of TCR repertoire in achalasia

Since T_RM presented dysregulated immune responses in achalasia, we next explored the TCR repertoire of T cells. We found more cloned expanded TCR repertoire in tissue than in blood and more clonal expansions in achalasia than in controls (Fig. 6a, Supplementary Data 7). The Gini index, positively associated with TCR clonality[43], was significantly higher in achalasia compared with that in controls (Fig. 6b). Shannon's entropy index, positively correlated with TCR diversity[43], was significantly lower in achalasia than that in controls (Fig. 6b). Collectively, these results indicated markedly clonal expansions but a decreased diversity of TCR repertoire of T cells in achalasia, in accordance with a previous study[44].

## Patient-specific TCR repertoire in achalasia

Owing to the clonal expansions in achalasia, we next calculated the degree of clonal overlap between samples. We found dominant clones overlapped between tissue and blood for the same patient but shared only at very low levels among different patients (Fig. 6c). Different subclusters of T cells had shared clones (Fig. 6d), indicating the same pseudotemporal cell trajectory. These results indicated clonal expansions of T cells were private to individual achalasia patients, in accordance with a previous study[44].

## A tendency of more human cytomegalovirus-specific CDR3 sequences in achalasia

Since a dysregulated immune response is hypothesized to be triggered by virus stimulation in achalasia[1], we then explored potential epitopes and related antigens by analyzing β-chain CDR3 sequences using VDJdb and TCRmatch methods. Although without statistical difference, we found a tendency of more human cytomegalovirus (CMV)-specific CDR3 sequences in achalasia than those in controls by using the VDJdb method (Fig. 6e). The statistically no difference of CMV might be caused by the limited numbers. Besides, the total virus-related sequences were more enriched in achalasia than those in controls, in expanded but not in non-expanded TCR repertoire (Supplementary Data 7). The top four prevalent virus epitopes enriched in patients were KLGGALQAK (CMV), GILGFVFTL (influenza A virus), NLVPMVATV (CMV) and FRCPRRFCF (CMV) (Supplementary Data 7). The findings of the VDJdb method were confirmed by the TCRmatch method (Fig. 6e). These results indicated CMV antigen might be a potential driver of T cell clonal expansions in achalasia.

## Clonal expansions of T_RM in achalasia

We next explored the TCR repertoire based on clusters. Although TCRαβ was detected in all T cell clusters except γδ T cells, cloned

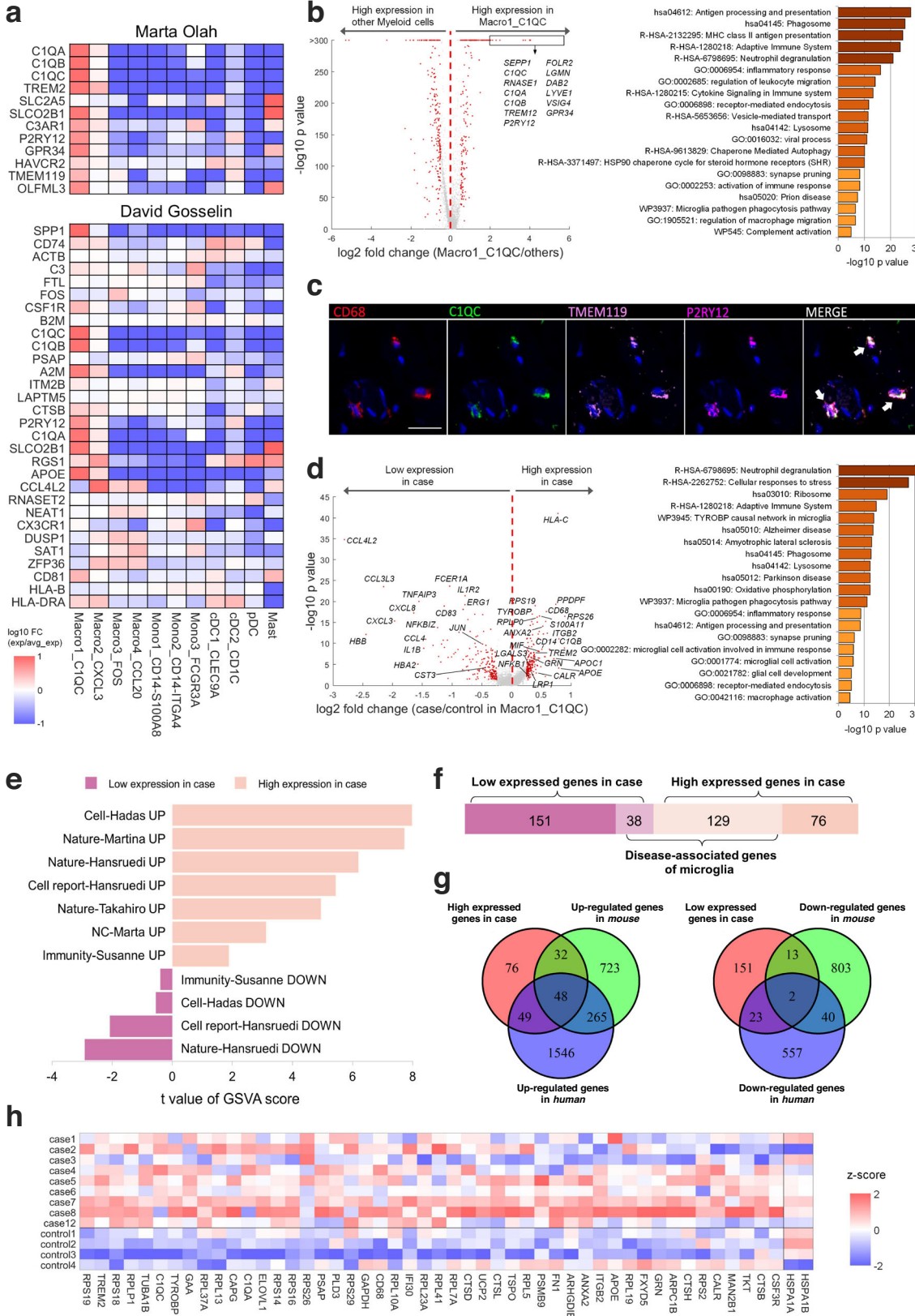

TCRαβ was mainly distributed in memory T cells and effector T cells (Supplementary Fig. 6a and Supplementary Data 7). We found $T_{RM}$ showed more expansions in achalasia than those in controls, especially in ZNF683⁺ CD8⁺ $T_{RM}$ and XCL1⁺ CD4⁺ $T_{RM}$ (Supplementary Fig. 6b and Supplementary Data 7). Differential TCR and VJ recombinations of tissue and blood are shown in Supplementary Fig. 6c.

We also compared expression levels of DEGs associated with cloned expanded and unexpanded T cells (Supplementary Fig. 6d). We found highly upregulated DEGs in expanded $T_{RM}$, effector T cells, and effector memory T cells, compared with unexpanded cells, such as *ZNF683*, *XCL2*, and *HLA-DRB1* in $T_{RM}$. We also found many pro-inflammatory factors in expanded effector T cells and effector memory T cells that were more

**Fig. 3 | The C1QC+ macrophages were transcriptionally similar to microglia and exhibited a neurodegenerative dysfunctional phenotype in achalasia.**
**a** Comparison of marker genes of myeloid clusters with previously identified human microglial gene sets reported by Marta Olah and David Gosselin. **b** Volcano plot (left) and Metascape analysis (right) of marker genes for C1QC+ macrophages compared with other myeloid clusters. **c** Co-localization of CD68 (red), C1QC (green), TMEM119 (pink) and P2RY12 (magenta) in C1QC+ macrophages of an achalasia patient. Scale bar, 20 μm. Arrow, C1QC+ macrophages. **d** Volcano plot (left) and Metascape analysis (right) of the DEGs for C1QC+ macrophages between achalasia and controls. **e** Gene set variation analysis (GSVA) analysis comparing DEGs for C1QC+ macrophages with signatures of neurodegenerative dysfunctional microglia. **f, g** Comparison of DEGs for C1QC+ macrophages with neurodegenerative dysfunctional genes of microglia by bar chart (**f**) and Venn diagram (**g**). **h** Heatmap showing the specific DEGs for C1QC+ macrophages that overlapped with neurodegenerative dysfunctional genes of microglia. Statistics: Volcano plots (**b**-left, **d**-left) were calculated by a two-sided Wilcoxon rank-sum test; Metascape analysis (**b**-right, **d**-right) was calculated by Fisher's exact test. Source data are provided as a Source Data file.

highly expressed in achalasia than in controls, including *TNF*, *IFNG*, *GZMB*, and *GZMH* (Supplementary Fig. 6d). These findings indicated the clonal expansions of T cells, especially $T_{RM}$, might be involved in the inflammatory process in achalasia.

### No significant clonal expansions of B cells in achalasia
Ninety-one percent of B cells showed no noteworthy clonal expansions. B cells with more than three BCR clones only existed in the blood of achalasia, mainly in case 11 (Supplementary Data 8). These findings indicated no significant clonal expansions of B cells in achalasia.

### DEGs for C1QC+ macrophages and $T_{RM}$ overlapped with risk genes for achalasia and other neurodegenerative diseases
We previously identified 66 common and rare gene variants associated with achalasia in immunologic and neurological genes by whole-exome sequencing (WES)[6], and the transcriptome of individual cell types determined here provided the expression levels of these genes. Here, we found that many of the WES-identified risk genes for achalasia were differentially expressed in most of the myeloid cells and $T_{RM}$ (Supplementary Fig. 7a). *HLAs* were differentially expressed in many types of immune cells, while other risk genes, such as *CUTA*, *SLC9A9*, and *MCOLN2*, were differentially expressed in specific types of immune cells. Thirteen genes associated with risk alleles of achalasia exhibited differential expression in C1QC+ macrophages, including *CUTA* and *HLA-DQB1* (Supplementary Fig. 7b), compared with 18 risk genes that overlapped with DEGs in $T_{RM}$, including *HLA-DRB1* and *HLA-DPB1* (Supplementary Fig. 7c). The *CUTA* allele was a common missense variant and was reproducibly associated with an increased risk of achalasia[6].

Comparison of the DEGs of immune cells in achalasia and risk genes of Genome-Wide Association Studies (GWAS) from other neurodegenerative diseases[45] showed a high correlation with PD and MS, especially MS (Supplementary Fig. 7d). Twenty-one DEGs for C1QC+ macrophages overlapped with risk genes for MS, which were mainly enriched in antigen processing and presentation by Metascape analysis, compared with 27 overlapped DEGs for $T_{RM}$, which were mainly enriched in regulation of leukocyte cell−cell adhesion and lymphocyte activation (Supplementary Fig. 8a−d and Supplementary Data 9). *HLA-B* and *HLA-DRB*, differentially expressed in C1QC+ macrophages or $T_{RM}$, were risk genes for both achalasia and MS (Supplementary Fig. 8a, b). Thus, the correlation between genetic risk factors and achalasia-specific transcription might partially explain the risk factor conferred by genetic variants.

### Achalasia was characterized by altered interactions between C1QC+ macrophages and $T_{RM}$
We explored achalasia-specific receptor-ligand interactions between myeloid cells and lymphocytes. Based on CellChat, we identified 7202 (3000 upregulated and 4202 downregulated) putative differential interactions in achalasia compared with controls, mainly in myeloid cells and $T_{RM}$ (Supplementary Fig. 9a and Supplementary Data 10). The most differential interactions were MHC-I, MHC-II and CCL (Supplementary Fig. 9b). Based on NicheNet, we confirmed differential ligand-to-receptor interactions between C1QC+ macrophages and $T_{RM}$ in achalasia compared with controls. Among these, the FN1 (C1QC+ macrophages)-ITGA4_ITGB1 complex ($T_{RM}$) pair was the most highly upregulated pathway (Supplementary Fig. 9c and Supplementary Data 10). By prioritizing differentially expressed ligands to differentially expressed target genes, we found $T_{RM}$ had more differentially expressed initiating ligands compared with C1QC+ macrophages. Among them, TNF, CCL, and HLA-DRA were the differentially expressed ligands for both C1QC+ macrophages and $T_{RM}$ (Supplementary Fig. 9d, e). Thus, we identified altered interactions between C1QC+ macrophages and $T_{RM}$ in achalasia.

### Increasing inflammation with achalasia progression
Using high-resolution manometry (HRM), achalasia can be classified as type I, II, or III, with type I and II accounting for more than 90% of cases. Previous studies indicated that type I might progress from type II and represent the late stage of achalasia[46−48]. We found that the key immune clusters, including C1QC+ macrophages, ZNF683+ CD8+ $T_{RM}$, and XCL1+ CD4+ $T_{RM}$, were higher in type I compared with type II patients ($n = 3$ for type I vs 6 for type II, Fig. 7a and Supplementary Data 11), indicating the higher inflammatory state of these clusters in type I. DEG analysis showed more upregulated genes in type I than in type II achalasia (Supplementary Fig. 10a, b). For C1QC+ macrophages, type I had higher expression of neurodegenerative dysfunctional genes, including *APOE*, *CD68*, and *TYROBP*, while type II had higher *C1Q*, *TMEM176B*, and *CRIP1* expression (Fig. 7b, c and Supplementary Data 11). Metascape analysis showed that C1QC+ macrophages of type I had greater enrichment for lysosome, phagosome, and cell activation, compared with type II enrichment for prion disease, C1Q complex, and mitochondrial oxidative phosphorylation (Fig. 7d). The difference was more apparent in the Metascape results for $T_{RM}$, which supported the higher T cell activation in type I achalasia (Fig. 7e). TCR clonal expansion of $T_{RM}$ was also greater in type I (case 5, 6 and 8) than type II (case 1−4, 7, 12) achalasia (Fig. 6a). Analysis of the pseudotemporal trajectories showed the pseudotime was earlier in type II than type I achalasia in most clusters, regardless of myeloid cells or lymphocytes (Fig. 7f and Supplementary Data 11). Analysis of a large database of 729 patients showed a longer disease duration for type I than type II achalasia (median duration, 5 years for type I ($n = 189$) vs 3 years for type II ($n = 540$); $P < 0.001$; Fig. 7g and Supplementary Data 11). These results suggested that type I progressed from type II, as demonstrated previously[46−48]. The immune cells were more activated in type I compared with type II achalasia, and the inflammation increased with disease progression.

### Achalasia-associated transcriptional changes of immune cells in peripheral blood
Immune cells in peripheral blood showed different transcriptomes from that in LES tissue, whereas the differential expressed genes between tissue and blood were quite similar among different cell types (Supplementary Fig. 11a and Supplementary Data 12). Heat shock proteins, tissue-specific marker of *CD69*, and pathways of IL-17, MAPK, and VEGFA-VEGFR2 were upregulated in LES tissue, while the WNT pathway was upregulated in blood (Supplementary Fig. 11a, b).

We explored achalasia-associated transcriptional changes of myeloid cells between achalasia and controls in peripheral blood even though with comparable compositions (Fig. 2b). Analysis of DEGs

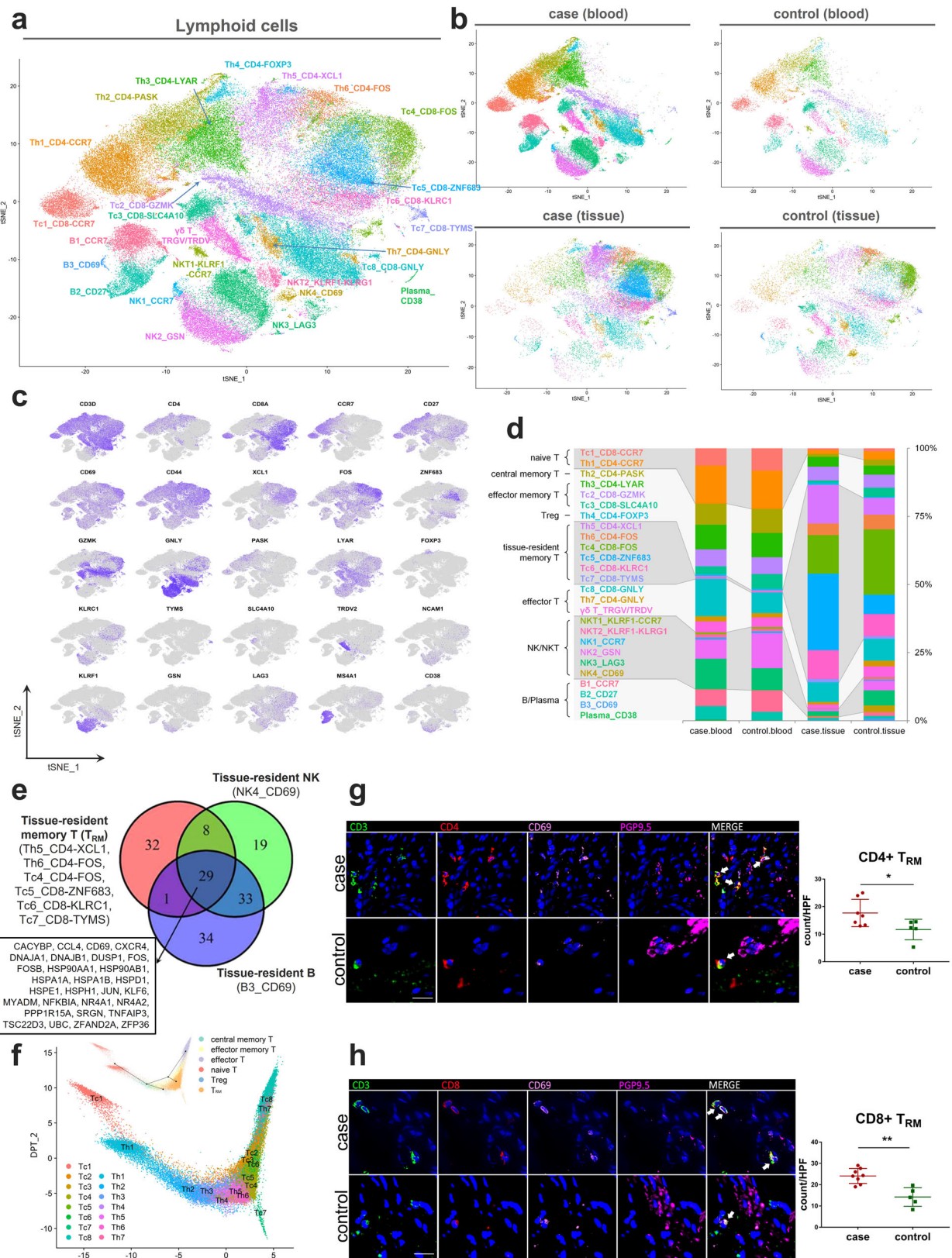

**Fig. 4 | Single-cell transcriptomics of subclustered lymphocytes. a**, **b** t-SNE plot of the 26 lymphocyte clusters (**a**) and detailed lineages in different tissue contexts (**b**). **c** t-SNE plots of canonical marker genes for 26 lymphocyte clusters. **d** Lymphocyte proportions for LES tissue and blood between achalasia and controls, colored by cell types. **e** Venn diagram showing the core signature shared by $T_{RM}$ and tissue-resident NK and B cells. **f** Pseudotemporal transcriptional trajectory of the T cells using the DPT and Slingshot algorithms. **g**, **h** Multicolor IHC and quantifications of CD4$^+$ $T_{RM}$ (**g**) and CD8$^+$ $T_{RM}$ (**h**) in the LES tissue between achalasia (CD4$^+$ $T_{RM}$, $n = 7$; CD8$^+$ $T_{RM}$, $n = 8$) and controls ($n = 5$). The $T_{RM}$ were increasingly infiltrated and surrounding the residual sparse myenteric plexus (shown by PGP9.5) in achalasia. Scale bar, 20 μm. Data are represented as mean ± SD. Statistics: two-tailed unpaired $t$-test. *$P < 0.05$; **$P < 0.01$. Arrow, CD4$^+$ and CD8$^+$ $T_{RM}$. Source data are provided as a Source Data file.

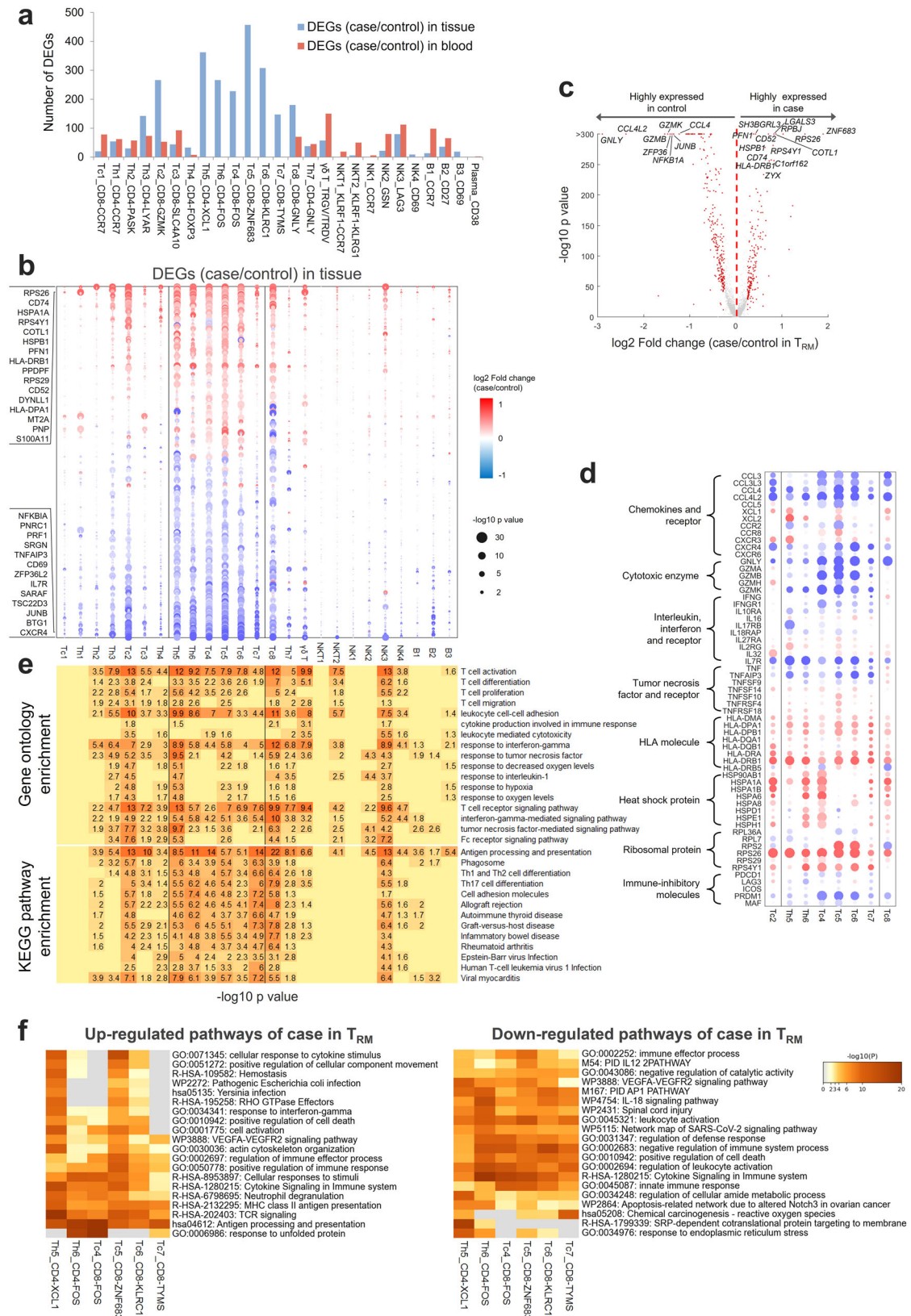

**Fig. 5 | $T_{RM}$ expressed transcripts involved in dysregulated immune responses in achalasia. a** A bar chart depicting the differentially expressed genes (DEGs) between achalasia and controls for the 26 lymphocyte clusters. **b** Dot plot of DEGs for the 26 lymphocyte clusters in blood and LES tissue. **c** Volcano plot showing the merged DEGs for $T_{RM}$. **d** Specific DEGs for $T_{RM}$, GZMK$^+$ CD8$^+$ T cells (Tc2), and

GNLY$^+$ CD8$^+$ T cells (Tc8). **e** GO (up) and KEGG (down) analysis of DEGs for the 26 lymphocyte clusters. **f** Metascape analysis of DEGs for $T_{RM}$. Statistics: $P$-value in (**b**–**d**): two-sided Wilcoxon rank-sum test; $P$-value in (**e**, **f**): Fisher's exact test. Source data are provided as a Source Data file.

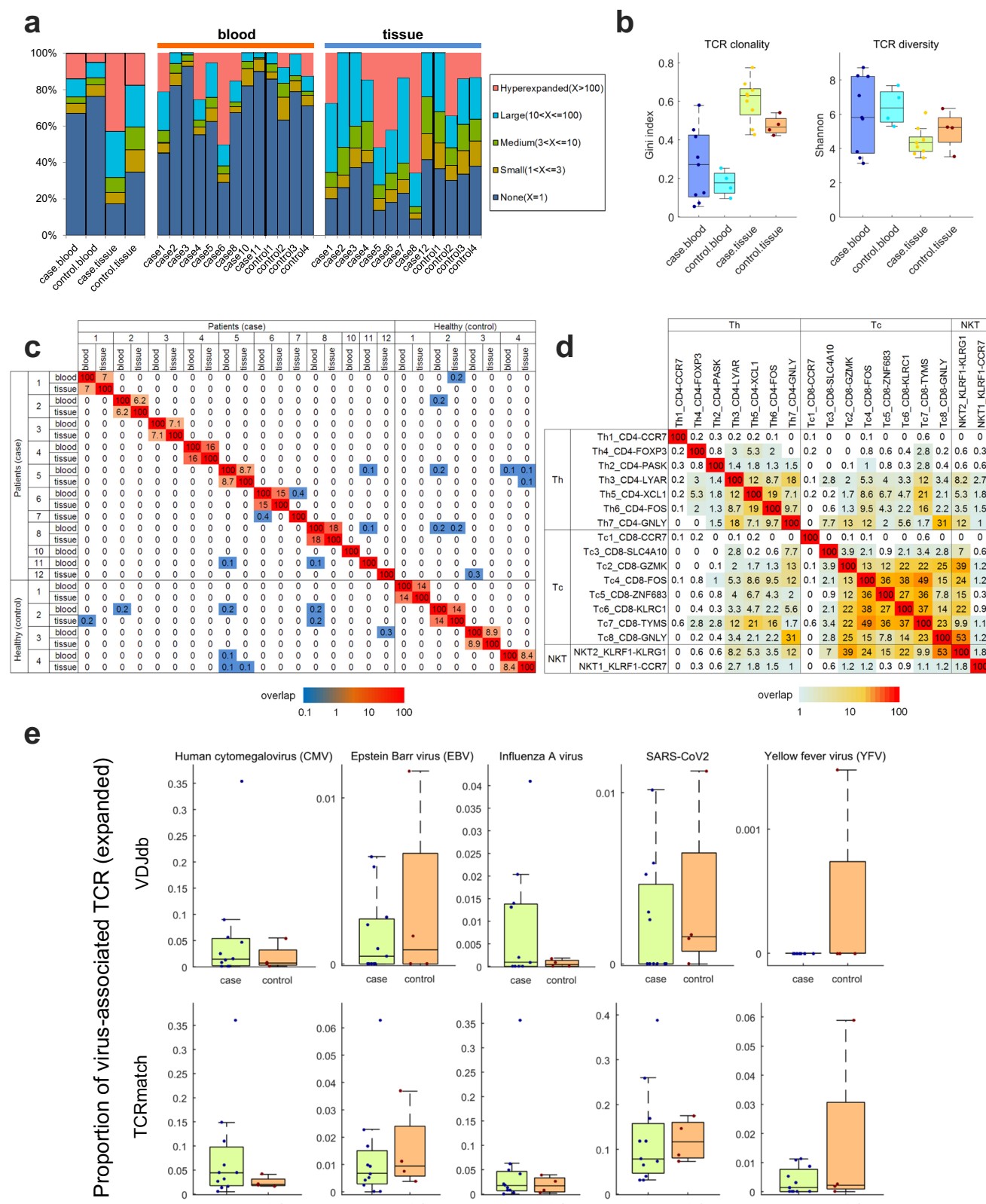

showed the upregulations of *PPDPF*, *NENF*, and *CYBA* and downregulations of *FOS*, *DUSP1*, and *GPX1* in peripheral blood myeloid cells of achalasia (Supplementary Fig. 12a and Supplementary Data 4). Metascape analysis showed increased expressed DEGs were most enriched in response of EIF2AK4 (GCN2) to amino acid deficiency, cellular response to cytokine stimulus, and type II interferon signaling (Supplementary Fig. 12a). Next, achalasia-associated transcriptional

changes of lymphocytes in peripheral blood were analyzed, especially for GNLY[+] CD8[+] T cells, CD27[+] B cells, and plasma, which were proportionally increased in achalasia (Fig. 4d). Analysis of DEGs showed the common upregulations of *SFPQ*, *KLF2*, and *CYBA* and common downregulations of *JUNB*, *FAM118A*, and *C21orf33* in peripheral blood lymphocytes of achalasia (Supplementary Fig. 12b and Supplementary Data 6). The most specific DEGs of GNLY[+] CD8[+] T cells were

**Fig. 6 | TCR repertoire in achalasia and controls. a** Comparison of the cloned TCR repertoire in blood and LES tissue between achalasia and controls. **b** Gini index and Shannon's entropy showed clonal expansions but a decreased diversity of TCR repertoire of T cells in achalasia. Case blood, *n* = 9; case tissue, *n* = 9; controls, *n* = 4. The results are depicted in boxplots, in which the value for each patient is represented by a dot, the upper and lower bounds represent the 75% and 25% percentiles, respectively, the center bars indicate the medians and the whiskers denote values up to 1.5 interquartile ranges above the 75% or below the 25% percentiles. **c** Limited

shared clones among different patients. **d** Abundant shared clones among different subclusters of T cells. **e** A tendency of more CMV-specific CDR3 sequences in achalasia than those in controls, calculated by VDJdb and TCRmatch methods. Case, *n* = 11; controls, *n* = 4. The results are depicted in boxplots, in which the value for each patient is represented by a dot, the upper and lower bounds represent the 75% and 25% percentiles, respectively, the center bars indicate the medians and the whiskers denote values up to 1.5 interquartile ranges above the 75% or below the 25% percentiles. Source data are provided as a Source Data file.

upregulations of *KLF2* and *PPDPF* and downregulations of *KIR2DL3* and *HLA-DRB5*. *CYBA* and *KLF2* were the common most upregulated DEGs of CD27⁺ B cells and plasma, while the most downregulated DEGs were *RPS29* and *JUNB* for CD27⁺ B cells and *IGKV5*-2 for plasma (Supplementary Fig. 12c). The most enriched pathways of GNLY⁺ CD8⁺ T cells, CD27⁺ B cells, and plasma were adaptive immune response, cellular response to cytokine stimulus, and rRNA processing in the nucleus and cytosol (Supplementary Fig. 12c).

## Discussion

In this study, we mapped the unbiased, comparative single-cell landscape of immune cells from peripheral blood and paired LES tissue in achalasia and controls. We identified the immune cells and transcriptomes in achalasia. The LES tissue of achalasia was enriched in C1QC⁺ macrophages and T_RM, especially around or even infiltrating the myenteric plexus of LES. This provides information on achalasia-associated immune cell subpopulations and their possible impact on the ENS, which was not previously reported.

We found that C1QC⁺ macrophages exhibited high expression of a previously identified microglial gene set[21,22] and Metascape analysis of marker genes of C1QC⁺ macrophages also enriched in the microglial pathogen phagocytosis pathway. Multicolor IHC and t-SNE analysis of CyTOF confirmed this C1QC⁺ macrophage cluster highly expressed C1QC, TMEM119, P2RY12, and TREM2, which are canonical markers of microglia. Microglia are the resident macrophages in the brain and are involved in the homeostasis, plasticity, immunity, and repair of the CNS[23]. Given the transcriptional similarity between C1QC⁺ macrophages and microglia and the wide distribution of ENS in the gastrointestinal tract, we speculated that C1QC⁺ macrophages might have an impact on the ENS.

We compared the DEG signature of C1QC⁺ macrophages from achalasia with the microglial signature of neurodegenerative diseases[9,10,21,29–32] and found many M0-homeostatic microglial transcripts were repressed in C1QC⁺ macrophages, while the key dysfunctional transcripts of microglia in neurodegenerative diseases were markedly upregulated. Among these, the upregulated APOE-TREM2 pathway was crucial for controlling the switch from a homeostatic to a neurodegenerative dysfunctional state of microglia during phagocytosis of apoptotic neurons. GSVA analysis also showed that DEGs for C1QC⁺ macrophages were similar to the neurodegenerative dysfunctional microglia and were supported by the KEGG analysis of C1QC⁺ macrophages. Thus, the transcriptional patterns of C1QC⁺ macrophages are similar to microglia and exhibit a neurodegenerative dysfunctional phenotype in achalasia, possibly leading to loss of immune cell homeostasis and neuronal damage. Since this study shows the importance of LES-infiltrated macrophages in the pathogenesis of achalasia, it is important to explore the mechanism of C1QC⁺ macrophages using further functional experiments in the future.

Although T cells widely infiltrate the myenteric plexus of LES in achalasia, it is not known whether CD8⁺ or CD4⁺ T cells predominate in LES tissue[49,50]. The specific achalasia-associated T cells in LES are unknown. The peripheral Th1, Th2, Th17, Th22, Treg cells, and Breg cells were reported to increase in achalasia[51,52]. However, previous studies only focused on the proportions of T cell subclusters by analyzing the IHC of LES tissue or flow cytometry of blood. The achalasia-specific key subclusters and transcriptional changes have not been

explored. Moreover, the circulating immune cells might not reflect the local immune state of LES.

We found T_RM were increasingly infiltrated in the LES tissue in achalasia and mainly restricted to the area surrounding the myenteric plexus, especially ZNF683⁺ CD8⁺ T_RM and XCL1⁺ CD4⁺ T_RM. T_RM are reported to provide key adaptive immune responses in multiple neurodegenerative diseases, including MS and PD[42,53]. We found the T_RM expressed dysregulated immune responses in achalasia, with some pro-inflammatory pathways upregulated while some others downregulated. We also found higher clonal expansions of T_RM in achalasia, which might be involved in the dysregulated inflammatory responses in achalasia.

In comparing the transcriptome of individual cell types with previously reported risk genes in achalasia and neurodegenerative disease risk genes, we found that WES-identified risk genes for achalasia and neurodegenerative disease risk genes were both widely expressed in most of the myeloid and T_RM clusters[6,45]. This indicates a correlation between genetic risk factors and achalasia-specific cell cluster transcription, which may partially explain the aberrant immune states conferred by genetic variants. Among these neurodegenerative diseases, risk genes of MS had strongest correlation with the DEGs of immune clusters of achalasia. Besides, the transcriptional phenotype of T_RM in achalasia was also partially consistent with that of MS[42]. These findings indicated an etiological similarity between achalasia and MS, and information gained about MS might apply to achalasia.

Previous studies indicated that type I achalasia might progress from type II and represent the late stage of achalasia[46–48]. HRM shows type II had more esophageal contractions compared with type I achalasia[46]. Besides, LES tissues of type I patients showed more severe aganglionosis, neuronal loss, and fibrosis[54,55]. The end stage of Type I might be the end stage of this disease. Myenteric neurons have marked depleted or completely disappeared in the LES tissue in patients with end-stage disease. As a consequence, no postdeglutitive contractility occurs in the esophagus, and the clinical manifestation is egaesophagus[56]. We found C1QC⁺ macrophages, ZNF683⁺ CD8⁺ T_RM and XCL1⁺ CD4⁺ T_RM in a higher proportion and a greater inflammatory state in type I compared with those in type II patients. The pseudotemporal trajectories showed the pseudotime was earlier in type II than type I achalasia in most clusters, and analysis of a large database of achalasia patients showed that type I had a longer disease duration than type II. These results indicated that type II progressed to type I achalasia in agreement with previous studies[46–48]. The immune cells were more activated in type I compared with type II achalasia, and inflammation increased with disease progression. We did not analyze type III achalasia because of the low incidence rate (0.03–1.63 per 100,000 persons per year)[1] and the rarity of type III (<10% of achalasia)[57].

There are several limitations of our study. First, we mainly found the transcriptional similarity between C1QC⁺ macrophages and microglia and showed transcriptional signatures of C1QC⁺ macrophages in achalasia were similar to those of disease-associated microglia from other neurodegenerative diseases, indicating C1QC⁺ macrophages exhibit a neurodegenerative dysfunctional phenotype in achalasia. However, no functional experiment was performed to verify the functional similarity between C1QC⁺ macrophages and microglia in our study. And no functional experiment was explored for the impact

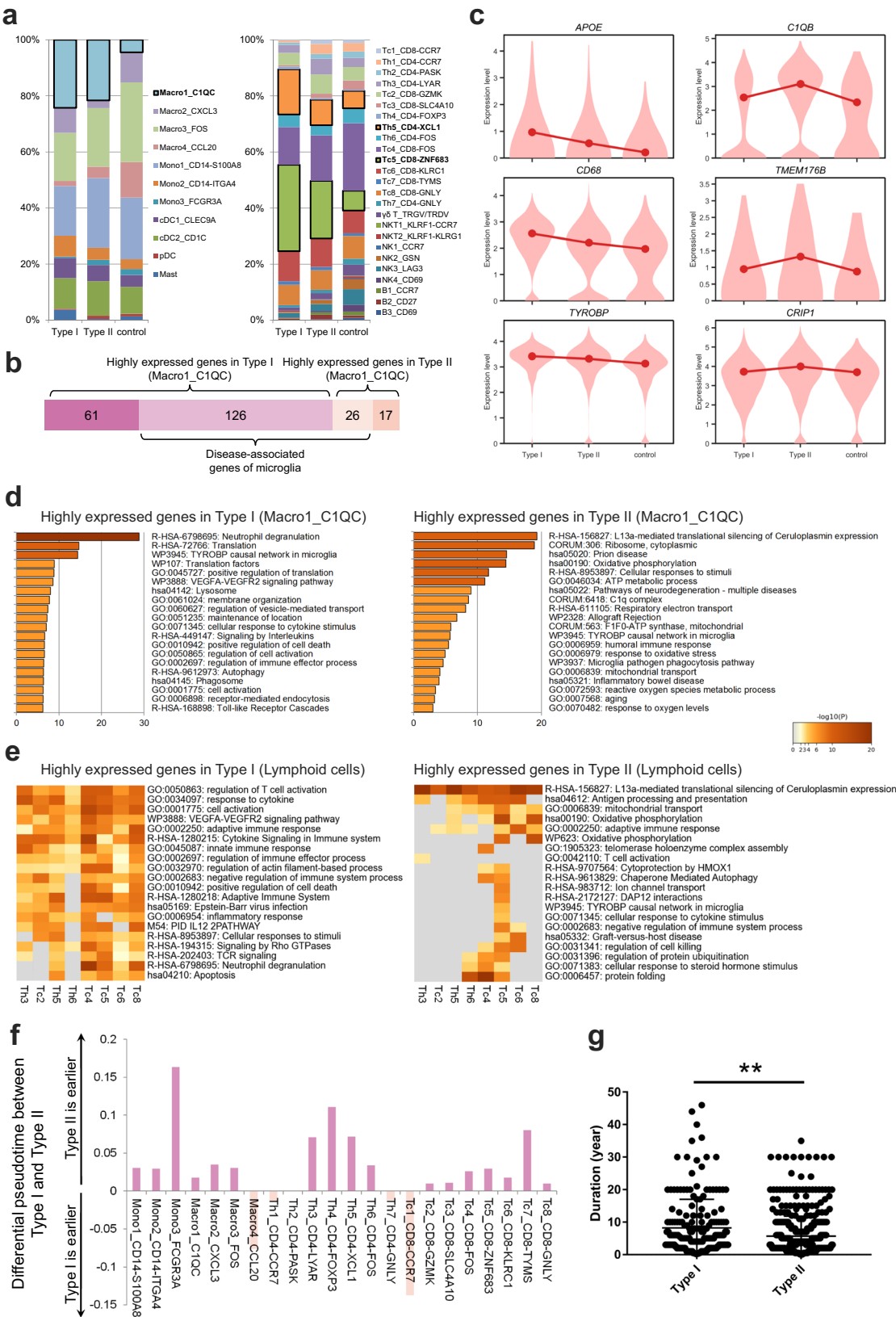

**Fig. 7 | Increasing inflammation with achalasia disease progression. a** Myeloid and lymphocyte cluster proportions for type I (*n* = 3) and type II (*n* = 6) achalasia and controls, colored by cell types. **b** Comparison of neurodegenerative dysfunctional genes of C1QC⁺ macrophages between type I and type II achalasia. **c** Specific neurodegenerative dysfunctional genes of C1QC⁺ macrophages in type I and type II achalasia and controls. **d, e** Metascape analysis of DEGs for C1QC⁺ macrophages (**d**) and T_RM (**e**) enriched in different pathways between type I and II achalasia.

Statistics: Fisher's exact test. **f** Pseudotemporal trajectory showed the pseudotime was earlier in type II than type I achalasia in most clusters, regardless of myeloid cells or lymphocytes. Data are represented as median. **g** Comparison of disease duration in type I (*n* = 189) and type II (*n* = 540) achalasia. Data are represented as mean ± SD. Statistics: two-tailed *t*-test. **P < 0.01. Source data are provided as a Source Data file.

of C1QC[+] macrophages on myenteric neurons. Additional investigation with functional study for C1QC[+] macrophages and $T_{RM}$ is needed to further explore the mechanism of the pathogenesis of achalasia in the future. Besides, we mainly validated the phenotype of $T_{RM}$ by multi-color IHC and CyTOF. However, we failed to validate the expressions of ZNF683 and XCL1 among $T_{RM}$ by multicolor IHC and CyTOF, which might be explained by the discrepant levels of RNA transcription and protein translation[58–60].

In summary, the transcriptional landscape of single immune cells from patients with achalasia and controls showed a disease-specific composition and transcriptional phenotype for immune cells in achalasia, especially in LES tissue, that might be involved in the inflammatory process in achalasia. C1QC[+] macrophages and $T_{RM}$ were significantly expanded and localized surrounding the myenteric plexus in the LES tissue of achalasia. C1QC[+] macrophages were transcriptionally similar to microglia of the CNS and exhibited a neurodegenerative dysfunctional phenotype in achalasia. $T_{RM}$ also expressed transcripts of dysregulated immune responses in achalasia. By potential interactions, the C1QC[+] macrophages, together with $T_{RM}$, might cause ganglionitis and loss of myenteric neurons. The loss of myenteric neurons could lead to aberrant esophageal peristalsis, dilated esophagus, impaired relaxation of the LES, and finally, the occurrence of achalasia (Supplementary Fig. 13).

## Methods

### Patient cohort and clinical characteristics

Eleven patients with idiopathic achalasia undergoing peroral endoscopic myotomy were enrolled at the Endoscopy Center and Endoscopy Research Institute, Zhongshan Hospital, Fudan University, Shanghai, China. Patients with achalasia were diagnosed by HRM, combined with history, gastroenterological endoscope, and barium esophagogram. HRM was evaluated by the Chicago Classification Criteria.

Clinical characteristics of the patients were collected prospectively by medical history questionnaires before the procedure. We obtained peripheral blood and paired LES tissues for nine achalasia patients and four controls and only peripheral blood for two achalasia patients. The peripheral blood was taken before the endoscopic procedure. LES smooth muscle specimens of achalasia patients were obtained during the peroral endoscopic myotomy. Control LES tissue specimens were taken from age-matched patients undergoing submucosal tunneling endoscopic resection for benign leiomyomas that originated from LES to ensure no invasion of the normal tissue. After the leiomyomas were removed, the control tissue specimens were taken from the surrounding normal tissue without tumor invasion. To reduce the impact of leiomyomas at maximum, we took the surrounding normal tissue 5 mm away from the leiomyoma. Exclusion criteria of participants included: cardiovascular, metabolic, hematologic, infectious, inflammatory, neoplastic or autoimmune disease, and history of surgery, radiotherapy or immunotherapy. Detailed clinical characteristics are summarized in Supplementary Data 1. Written informed consent was provided by all participants. This study was approved by the Research and Ethical Committee of Zhongshan Hospital, Fudan University, and complied with all relevant ethical regulations.

### Multicolor IHC

The formalin-fixed-paraffin-embedded sections were stained using Opal Multicolor Manual IHC Kit (Catalog # RC0086-56, Recordbio, Shanghai, China) according to the manufacturer's protocol, as previously described[61]. In brief, after sections were deparaffinized and rehydrated, the antigen was retrieved by immersing in boiling AR9 buffer in the oven for 15 min. After blocking with serum at room temperature for 10 min, the sections were incubated with primary antibodies overnight at 4 °C. Detailed information on primary antibodies is shown in Supplementary Data 13. The secondary horseradish peroxidase-conjugated antibody (HRP anti-Rabbit IgG antibody, Catalog # ab288151, Abcam, Cambridge, UK) was added with 1: 200

dilution and incubated for 50 min at room temperature. Signal amplification was performed by Tyramide signal amplification (TSA) working solution and incubated at room temperature for 10 min. After incubation with fluorescent dye-TSA solution, the sections were treated with boiling Ethylenediaminetetraacetic acid (EDTA [Catalog # 60-00-4, Sigma-Aldrich, Missouri, USA]) antigen retrieval buffer in the microwave. The next primary antibody was added and incubated overnight. After the sections were treated with all primary and secondary antibodies, the DAPI solution was added, and the sections were incubated at room temperature for 10 min. After incubation with spontaneous fluorescence quenching reagent for 5 min, the sections were covered with an anti-fade mounting medium. Finally, the multi-spectral imaging was collected based on the appropriate wavelength by Pannoramic Digital Slide Scanners and analyzed by Pannoramic Scanner (Akoya Biosciences, California, USA). Statistics was performed using GraphPad Prism (v8).

### Mass spectrometry

Tissue was washed and cut into slices in Phosphate buffer saline (PBS [Catalog # 70011036, Thermo Fisher, Massachusetts, USA]), 200 µl radioimmunoprecipitation assay (RIPA) lysis buffer (Catalog # R0278, Sigma-Aldrich) (containing 0.1% PMSF [Catalog # 329-98-6, Sigma-Aldrich]) was added, the sample was ground and then sonicated for 5 min, and lysed on ice for 30 min. Afterward, the sample was centrifuged at 13523×$g$ for 15 min. The supernatant was collected, and protein was measured by the bicinchoninic acid (Catalog # B9643, Sigma-Aldrich) method. Protein (200 µg) was dissolved in RIPA lysis buffer and reduced with 10 mM Dithiothreitol (Catalog # 3483-12-3, Sigma-Aldrich) at 37 °C for 1 h and alkylated using 20 mM iodoacetamide (Catalog # 144-48-9, Sigma-Aldrich) in the dark at room temperature for 30 min. Four volumes of cold acetone were added, and proteins were precipitated at −80 °C overnight. The solution was centrifuged at 13523×$g$ for 15 min to remove the supernatant, and the pellet was dissolved in 200 µl 50 mM $NH_4HCO_3$. Trypsin (Catalog # T6567, Sigma-Aldrich) (trypsin: protein = 1: 50, w/w) was added and incubated at 37 °C overnight. Formic acid (Catalog # 64-18-6, Sigma-Aldrich) at 5% was added to terminate digestion. Finally, the sample was desalted using a desalting column (Waters) for mass spectrometry analysis.

Mass spectrometry analysis was performed on a TIMS-TOF Pro mass spectrometer (Bruker Daltonics, Massachusetts, USA) equipped with a nano-electrospray ion source. Full-scan mass spectrometry spectra (m/z 100–1700) were acquired with iron mobility ranging from 0.7–1.3 V/cm$^2$. Tandem mass spectrometry acquisition was performed within a 1.16 s cycle time. The intensity threshold was 5000, while accumulation and release time was 100 ms. The electrospray voltage was 1500 V, and the auxiliary gas was set at 3 L/min. The ion source temperature was set at 180 °C.

Tandem mass spectra were analyzed using PEAKS Online (Bioinformatics Solutions Inc, Ontario, CA). Raw data files were searched against the human database from Swiss-Prot using a 15-ppm precursor mass tolerance and a 0.05 Da fragment mass tolerance. Carbamidomethylation on cysteine was set as a fixed modification. Acetylation on protein N-terminal, oxidation on methionine, and deamidation on asparagine were set as variable modifications. Protein peak areas were searched for further statistical analysis.

### CyTOF sample preparation

The LES tissue specimens for achalasia and controls were taken from three patients of achalasia and three age-matched patients with benign leiomyomas, the same as those for scRNA-seq. The freshly collected tissue samples were washed twice with $Ca^{2+}$- and $Mg^{2+}$- Hanks' Balanced Salt Solution (HBSS [Catalog # H9394, Sigma-Aldrich]) at 4 °C and then transferred into digestion medium of $Ca^{2+}$- and $Mg^{2+}$- HBSS, fetal bovine serum (FBS [Catalog # 10099, Sigma-Aldrich]) 1%, DNase I 0.5 mg/ml (Catalog # 11284932001, Sigma-Aldrich), Collagenase IV

0.5 mg/ml (Catalog # C4-BIOC, Sigma-Aldrich). After mincing into 1-mm³ pieces, the samples were digested for 40 min at 37 °C under gentle agitation. After digestion, the dissociated cells were filtered through a 70-μm cell strainer, washed in PBS (1% FBS), and centrifuged at 300×g for 5 min. The supernatant was removed, and the pellets were suspended in red blood cell lysis buffer (Catalog # 420301, BioLegend, California, USA) and incubated on ice for 2 min. After red blood cell lysis, cells were washed and resuspended as before.

The dissociated cells were stained with 100 μl Living Cell Staining Mix (Catalog # 201194, Fluidigm, California, USA) for 5 min on ice. The samples were blocked and stained with a surface antibody panel for 30 min and followed by fixation overnight. Perm Buffer (Catalog # GAS004, eBioscience, California, USA) was utilized before intracellular marker staining according to the manufacturer's instructions. Cells were then washed, resuspended, and applied to a mass cytometer (Helios, Fluidigm). Detailed antibody information is listed in Supplementary Data 13.

## CyTOF data analysis
Mass cytometry data of each sample were first de-barcoded using a doublet-filtering scheme with mass-tagged barcodes. The FlowJo software (Version 10.8.1, FlowJo, Oregon, USA) was used to gate-retain live, singlet, valid immune cells. Batch effects from different samples were normalized using the bead normalization method. Separate t-SNE plots of all CD45[+] cells were developed by FlowJo software to annotate specific clusters. Statistics was performed by GraphPad Prism (v8).

## Single-cell collection for scRNA-seq
The LES tissue samples were digested for achalasia and controls with the same protocol as CyTOF. After digestion, the dissociated cells were isolated by gradient centrifugation through Percoll solution (Catalog # 17-0891-09, GE Healthcare, Sweden) to acquire LES muscle mononuclear cells as described previously[44,62,63]. Percoll solutions (44% and 67%) were prepared by diluting the stock solution (45 ml Percoll and 5 ml of 10× PBS) in an R-10 medium. Cells were first resuspended in the 44% Percoll solution, and 6 ml of this cell suspension was gently layered on 3 ml of 67% Percoll solution. The mixed suspension was then centrifuged at 600×g for 20 min. Mononuclear cells that migrated to the interface were carefully removed and washed twice in PBS (1% FBS). The final cells were then resuspended in PBS.

PBMCs were isolated using Histopaque-1077 (Catalog # 10771, Sigma-Aldrich) per the manufacturer's instructions. Briefly, 4 ml of fresh peripheral blood was collected in EDTA anticoagulant tubes and layered onto Histopaque-1077. After centrifugation, enriched lymphocyte cells at the plasma-Histopaque-1077 interface were collected and washed twice with PBS. These mononuclear cells were resuspended in PBS. The viability of the final cell suspension was calculated using a hemocytometer with trypan blue (Catalog # 25-900-CI Corning, New York, USA). Since both tissue and blood cells were isolated without using fluorescence-activated cell sorting, cell viability was maximized (mostly about 95% in tissue and 98% in PBMC). The PBMC and LES muscle mononuclear cells were separately sampled and disassociated without mixing.

## Single-cell library construction and sequencing
Single-cell suspensions of LES tissue and blood were separately loaded on the Chromium System without mixing using the Chromium Single Cell 5' Library & Gel Bead Kit v3 (10X Genomics, California, USA). Following capture and lysis, single-cell gene expression and libraries were constructed according to the manufacturer's instructions. Completed libraries were sequenced on NovaSeq6000 (Illumina, California, USA) platforms using 2 × 150 chemistry at a targeted median read depth of about 63,000 reads per cell from total gene expression libraries and about 13,000 reads per cell for TCR libraries (cycle specifications 150:8:0:150 [R1:i7:i5:R2]).

## Single-cell raw data processing
Illumina basecall files (*.bcl) were converted to fastqs using the *Cell Ranger* v3.0.1 pipeline with recommended parameters. FASTQ files were then aligned to the GRCh38 human reference genome to generate the gene-cell unique molecular identifier (UMI) matrix. This output matrix was then imported into the Seurat (version 4.0.2) R toolkit for quality control and downstream analysis. Low-quality cells with the following criteria were excluded: (1) <1000 UMIs; (2) <500 genes; or (3) a mitochondrial genome detection rate >10%. The sequencing information for genes and cells for each individual in the study is shown in Supplementary Data 1.

## Batch effect correction and normalization
We performed batch correction across 28 samples in 10X Genomics scRNA-seq data by the fastMNN method (https://MarioniLab.github.io/FurtherMNN2018/theory/description.html), which is a faster and more robust version of the MNN method[64]. This method performs a multi-sample principal component analysis (PCA) and subsequently performs all calculations in the principal component (PC) space instead of the original gene space. This optimizes the computation speed and reduces the noise in neighbor detection. We applied the R package *batchelor* (version 1.6.2) to perform fastMNN and the number of PCs was set as 80, and the number of nearest neighbors in the same batch *knn* was set as 50. The corrected output is a matrix of samples × PCs and was used as the input for further clustering, visualization, and pseudo-trajectory analyses.

## Cell doublet detection and removal
We used a cluster-level approach to remove the doublet clusters that contained potential doublet cells[12]. Specifically, we removed the CD14[+] IL32[+] cluster and FCGR3A[+] IL32[+] cluster with a large fraction of potential monocyte-T cell doublets, which expressed both monocyte signature genes (*CD14*, *FCGR3A*, *S100A8*, *S100A9*) and T cell signature genes (*IL32*, *CD3D*, *CD3E*).

## Unsupervised clustering analysis
We used an unsupervised graph-based clustering algorithm implemented in Seurat 4 (version 4.0.2) to cluster single cells by their expressions[65]. We performed fastMNN to correct batch effects and then used the function *FindNeighbors* and *FindClusters* on the corrected 80 PCs with a resolution of 0.5-2 to cluster all single cells. Known markers were used to annotate major cell types, including immune cells (lymphocytes and myeloid cells) and non-immune cells (smooth muscle cells, endotheliums, fibroblasts, platelets and HSPCs). We selected myeloid cells and lymphocytes for second-round clustering with the same procedure for further analysis.

## Dimensionality-reduction and visualization using t-SNE
For dimension reduction and visualization, we used the function *RunTSNE* in Seurat to perform t-SNE (t-distributed stochastic neighbor embedding). The PCs used to calculate the embedding were the same as those used for clustering.

## Identification of differentially expressed genes
We identified DEGs based on the Wilcoxon rank-sum test that was implemented in the function *FindMarkers* and *FindAllMarkers* in the Seurat package. Unless noted otherwise, we selected the genes with $P < 0.05$, absolute log2 fold change (after Laplace transformation) >0.25, and minimum fraction >0.1 as DEGs.

## Gene set variation analysis
We used R package *GSVA* (version 1.38.2) to perform gene set variation analysis[66]. GSVA assesses the relative enrichment of gene sets across samples using a non-parametric approach. Conceptually, GSVA transforms a gene expression matrix with p genes by n samples into a

pathway enrichment matrix with g gene sets by n samples. This facilitates many forms of statistical analysis of pathways or gene sets rather than genes, such as the comparison of pathway difference between case and control, which provides a higher level of interpretability. The pathway or gene set difference between case and control was calculated based on a two-sample *t*-test, and the positive value of the *t*-score represents the gene set was upregulated in the case, while the negative value of *t*-score represents the gene set was downregulated in the case.

## M1/M2 macrophages definition

The M1/M2 phenotype of each macrophage cell type was defined as the mean expression of the gene signatures[12]. Genes associated with M1 macrophages include *CCL5, CCR7, CD40, CD86, CXCL9, CXCL10, CXCL11, IDO1, IL1A, IL1B, IL6, IRF1, IRF5* and *KYNU*, while *CCL4, CCL13, CCL18, CCL20, CCL22, CD276, CLEC7A, CTSA, CTSB, CTSC, CTSD, FN1, IL4R, IRF4, LYVE1, MMP9, MMP14, MMP19, MSR1, TGFB1, TGFB2, TGFB3, TNFSF8, TNFSF12, VEGFA, VEGFB* and *VEGFC* were used to define M2 macrophages.

## Pseudo-trajectory inference

We used three algorithms based on different principles to construct pseudo-trajectories. (1) Diffusion pseudotime (DPT), a method based on diffusion map, which measures transitions between cells using diffusion-like random walks[67]; (2) Potential of Heat diffusion for Affinity-based Transition Embedding (PHATE), a visualization method that captures both local and global nonlinear structure using an information-geometric distance between data points[68]; (3) Monocle 2, a method using reversed graph embedding to describe multiple fate decisions in a fully unsupervised manner[69]. MATLAB package *DPT* (version 1.0), R package *phateR* (version 1.0.7) and *monocle* (version 2.4.0) was used to perform this analysis, respectively, in which the input data was the corrected 80 PCs from fastMNN. The parameters of DPT were *transition matrix construction method* = 'nearest neighbors', *knn* = 20, *gstatmin* = 1.01, *nsig* = 10. The parameters of PHATE were *knn* = 20, *decay* = 10. The parameters of Monocle 2 were all default parameters. In addition, we also used Slingshot to validate the cell lineages of myeloid cells, which was performed by R package *slingshot* (version 2.4.0) with all default parameters[70].

## TCR and BCR analysis

The single-cell TCR-seq and BCR-seq were performed on the same cells as single-cell RNA-seq, and thus, we could get not only the transcriptome of each single cell but also its TCR or BCR simultaneously. The single-cell library construction and sequencing were performed simultaneously for scRNA-seq and scTCR(BCR)-seq, which had been described before. After sequencing, the raw data of TCR(BCR)-seq for 28 samples was further processed using *Cell Ranger* (version 3.0.1) against the human VDJ reference genome, and TCR(BCR) amino acid sequences were artificially translated from their nucleic acid sequences. Unavailable cells were filtered, including the cells with NA value in any chain or the cells with VJ but without CDR3, and only the cells with one α-β pair were retained. After filtration, we found the number of T cells in case7.blood and case12.blood was quite low (0 and 5 cells, respectively), so we removed the two samples in TCR analysis. At last, 52,649 T cells and 7716 B cells were retained after filtration.

In the TCR analysis, if two or more cells had identical α-β pairs and identical CDR3 amino acid sequences, these α-β pairs were identified as cloned TCRs, and the corresponding T cells were identified as cloned T cells. We calculated the Gini index to measure the equality of TCR distribution, which was positively correlated with TCR clonality[43]. We also calculated Shannon's entropy index to evaluate the TCR diversity of each sample[43]. Both the Gini index and Shannon entropy index were calculated by the R package of *tCR* (version 3.13). In

addition, we calculated the overlap of clones shared between two repertoires *X* and *Y*: overlap $(X, Y) = |X \cap Y| / \min(|X|, |Y|) * 100$, where $|X|$ and $|Y|$ are the clonal sizes (number of unique clones) of repertoires *X* and *Y*.

To identify the antigen epitopes and related viruses, we inputted β-chain CDR3 sequences of achalasia patients and controls into both VDJdb[71] and TCRmatch[72] methods. VDJdb (https://vdjdb.cdr3.net/) is a curated database of TCR sequences with known antigen specificities, which facilitates access to existing information on TCR antigen specificities. We first downloaded the database, including the CDR3 sequences and their corresponding VJ combinations, epitopes, epitope genes and epitope species. Then, we compared the β-chain CDR3 sequences and VJ combinations of our TCR data with the database to identify which epitopes and which microbes were more associated with achalasia based on the comparison between case and control. Furthermore, to validate the results of VDJdb, we inputted β-chain CDR3 sequences of cloned TCR into TCRmatch (http://tools.iedb.org/tcrmatch/), a tool of predicting TCR specificity by using a comprehensive k-mer matching approach to identify similar sequences that were previously characterized in the Immune Epitope Database[73].

To integrate TCR results with the gene expression, we mapped the cells with TCR to clustering results and identified 47,045 T cells with TCRαβ, including 20,780 CD4+ T cells, 25,641 CD8+ T cells, and 624 NKT cells. The distribution of TCR and cloned TCR in each cell type is presented in Supplementary Data 7. And further differential expression analysis between cloned T cells and non-cloned T cells was based on the 47,045 cells.

## Enrichment analysis for signature genes

We used the function *enrichGO* and *enrichKEGG* implemented in R package *clusterProfiler* (version 3.18.1) to perform GO and KEGG enrichment analysis for signature genes[74]. The genome-wide annotation for humans was downloaded from R package *org.Hs.eg.db* (version 3.14). The KEGG pathway information was from http://www.genome.jp/kegg/catalog/org_list.html. In addition, we also used *Metascape* (https://metascape.org) to perform the enrichment analysis for multiple gene lists[75].

## Comparison with other neurodegenerative diseases

We compared the DEGs in our dataset with the risk genes for other neurodegenerative diseases from the GWAS catalog (https://www.ebi.ac.uk/gwas/)[45], including PD, AD, amyotrophic lateral sclerosis (ALS), and MS. The strength of the association between achalasia and other neurodegenerative diseases was positively correlated with the number of genes located in the intersection of the two gene sets. Fisher's exact test was used to calculate the statistical significance.

## Cell−cell communication analysis

Based on the ligand−receptor interaction database curated in CellChat (version 1.1.0), we calculated the upregulated pathways and downregulated (case/control) pathways between different cell types[76]. The upregulated pathways in case were defined as significantly upregulated of ligand ($P < 10^{-6}$, log2FC > 0.5, minimum fraction > 0.1) and no significantly downregulated of receptor (log2FC > −0.2), or significantly upregulated of receptor ($P < 10^{-6}$, log2FC > 0.5, minimum fraction > 0.1) and no significantly downregulated of ligand (log2FC > −0.2). The downregulated pathways in case were defined as significantly downregulated of ligand ($P < 10^{-6}$, log2FC < −0.5, minimum fraction > 0.1) and no significantly upregulated of receptor (log2FC < 0.2), or significantly downregulated of receptor ($P < 10^{-6}$, log2FC < −0.5, minimum fraction > 0.1) and no significantly upregulated of ligand (log2FC < 0.2).

We also performed a NicheNet analysis on the interaction between C1QC+ macrophages and $T_{RM}$. NicheNet's prior model goes

beyond ligand–receptor interactions and incorporates intracellular signaling as well; thus NicheNet can predict which ligands influence the expression in another cell and which target genes are affected by each ligand[77]. The list of prioritized ligands for each 'sender' cell type was first identified based on the top differential expressed genes (case/control) that were found in the 'receiver' cell type and were further selected based on the differential expression of ligands themselves. R package nichenetr (version 1.1.0) was used to perform NicheNet analysis, and the code was deposited in GitHub (https://github.com/saeyslab/nichenetr).

### Reporting summary

Further information on research design is available in the Nature Portfolio Reporting Summary linked to this article.

## Data availability

Single-cell RNA-seq, TCR-seq, and BCR-seq data generated in this study have been deposited in the Genome Sequence Archive for Human (GSA-Human) database of the National Genomics Data Center under the primary accession code HRA002791. There are no restrictions on data availability. All other relevant data supporting the findings of this study are available within the article and its Supplementary Information or Source Data files or from the corresponding authors upon request. Source data are provided with this paper.

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

## Acknowledgements

We thank all the patients who contributed to this study and the generous support of our endoscopy teams, who made this work possible. We acknowledge the support of Weiwei Yang, Yi Zhang, Jiawei Li, and Zhening Pu for the project designing, data analyzing, and manuscript revising. This study was supported by grants from the Natural Science Foundation of China (82170555 [Q.L.], 82000507 [Z.L.], 62132015 [H.D.], 81873552 [Q.L.], 81670483 [P.Z.], 32100508 [H.D.], 31930022 [L.C.], 12131020 [L.C.], and T2341007 [L.C.]), Shanghai Collaborative Innovation Center of Endoscopy (P.Z.), National Key R&D Program of China (2022YFA1004800) (L.C.), Shanghai Rising-Star Program (19QA14 01900) (Q.L.), Yangfan Program of Shanghai Municipal Science and Technology Committee (20YF1406400) (Z.L.), Major Project of Shanghai Municipal Science and Technology Committee (18ZR1406700 [Q.L.] and 19441905200 [W.C.]), Strategic Priority Research Program of the Chinese Academy of Sciences (XDB38040400) (L.C.), and JST Moonshot R&D (JPMJMS2021) (L.C.). Supplementary Figs. 1a and 13 were created with BioRender.com.

## Author contributions

Q.L., P.Z., and L.C. conceived the study concept and design. Z.L., H.D., L.Y., W.C., Y.W., L.M., X.L., S.L., M.H., P.G., X.L., X.X., J.X., K.W., and L.W. contributed to the literature search, data interpretation, sample collection, and data processing. Z.L. and L.Y. did most of the experiments. Z.L. and H.D. made figures and tables. Z.L. and H.D. wrote the initial manuscript. All authors contributed to reviewing and editing the manuscript.

## Competing interests

The authors declare no competing interests.
