## [Peer Review File · Nature Communications]

A Single-Cell Transcriptional Landscape of Immune Cells in Human AchalasiaREVIEWER COMMENTS

Reviewer #1 (Remarks to the Author):

The manuscript by Liu et al profiles the LES and blood immune cell transcriptional landscape in achalasia using single-cell RNAseq. The authors identify both cellular and transcriptional features associated with achalasia, including increased frequencies of C1QC+ macrophages and inflammatory Trm cells, especially ZNF683+ CD8+ and XCL+ CD4 Trms. They further suggest that the C1QC+ achalasia enriched macrophage share gene expression profiles with microglia and have features similar to a number of neurodegenerative diseases including MS. The study also identifies clonal enrichment of T cells in cases with achalasia and proposes that type I achalasia progresses from type II by the nature of having a more inflammatory signature.

This study provides an excellent resource of single-cell RNAseq dataset of gene expression profiles and TCR/BCR profiles in blood and LES tissue of achalasia and control patients and identifies perhaps key immune features of the disease. However, there are a number of issues including lack of validation of the key findings and over interpretation of the data that detract from the manuscript. Additionally, the organization of the manuscript (general characterizing of LES/blood immune cell and comparison between disease and control are back and forth) and grammatical errors make it challenging to interpret at times. There are also some missed opportunities with their datasets that should be explored further including more detailed TCR analysis and blood signature analysis. Finally, the figures would benefit from increased resolution and improved labeling of data.

Major points:

1. At a minimum, the two key findings of C1QC+ macrophages and ZNF683+ CD8+ and XCL+ CD4 Trms need to be validated. The current IHC data in the paper is not specific to these cell types and not very convincing.
2. The authors presume functionality from transcriptional data which is not directly possible. For example, the authors state the C1QC+ macrophages are functionally similar to microglia. Functional experiments would need to be performed to make such conclusions. Currently, the cell types can only be claimed to have similar transcriptional signatures.
3. The increase in TCR clonality is interesting but the data is only shallowly interpreted. What are the clones, are they public, private? Are there shared clones between patients, between tissue and blood. The manuscript would benefit from a much more in-depth TCR investigation.
4. The blood data is very superficially analyzed throughout the manuscript. Having a biomarker for achalasia and particularly more severe achalasia would potentially be of benefit. However, little is mentioned about the peripheral blood signatures unique to achalasia throughout the manuscript. A more in-depth analysis would be of benefit here.
5. The manuscript would benefit from a thematic re-organization, improved resolution and labeling of figures and correction of grammatical errors throughout.

Minor points:

1. A number of statements are made about data throughout the manuscript that are not shown in the figures. All data needs to be shown in the figures. Some examples:
 - a. In the Single-cell transcriptomics of subclustered myeloid cells section: "We found expression of CD4 and TIM4 in the C1QC+ and CXCL3+ macrophages cluster", however there are no figures showing CD4 and TIM4 macrophage expression.
 - b. A number of genes that the authors refer to are missing from the figures throughout the manuscript
2. The IHC data presented in the manuscript is hard to interpret as it shows only small areas with unconvincing staining. For example:
 - a. Fig 1f-g authors state that macrophages (CD68) were increased in achalasia. However, only 1 cell is positive in both the control and achalasia image that is shown. The ganglia staining (PGP9.5) is over-exposed in controls and hard to see in the case that is shown. So tricky to interpret their data to be infiltration of immune cells into the ganglion. From the image shown only the increase in CD4/8 in the achalasia image is convincing. However, neither CD8 or CD4 staining by itself can be used for

- definitive T cell identification as both markers are expressed in other cell types. CD4 for examples is expressed on intestinal macrophages (as the authors point out) and CD8 is expressed in NK cells.
- b. Fig 3b only shows images of an achalasia case without any control images and would need to show the comparison to validate increase in C1qc macrophages in achalasia. Trem2 and C1Qc staining is remarkably similar suggesting that one is perhaps the background of the other.
 - c. Presence of the macrophages near the ganglion cells does not by themselves suggest that the pathophysiology of the disease is neurodegeneration.
 - d. Figure 4g. CD103 is only on 1 cell. CD8/CD4 staining even in combination with CD69 (which is not on all the cells) does not confirm that these cells are TRMs. Again no data on control vs achalasia here
3. There are numerous sentences throughout the manuscript that are challenging to interpret and should be restated. For example:
 - a. "We found the functional homogeneity but more heterogeneity among myeloid cells in achalasia"? Hard to understand as they're only showing the transcriptional profiles, no functional assay.
 4. Figure 3 needs to be quantified.
 5. No graph in main Figure 4 showing the marker genes used to identify different T cell types.
 6. Figure 4e is poorly described. From the pseudotime trajectory, most Trms are in transitional stages in naïve cell differentiating to effector memory T cells pathway. Tc7 cells were branch out, suggesting the unique features (cycling genes) compared to other T cells.
 7. DEGs in Figure 5 are these for Trm in achalasia VS Trm in controls or Trm VS other T cells in achalasia?
 8. Figure 5. f only one pathway analysis is shown is this KEGG? GO? Unclear and both are stated in the text. g, e the conclusion that these cells are pro-inflammatory is not supported fully by the data as a number of inflammatory cytokines were not upregulated in these clusters.
 9. Figure 6 e. the sentence that clonal expansion is evidence of pathogenesis is not correct. One can have expanded T cells clones that do not cause disease.
 10. Unclear how to interpret data in Figure 7f as type I and type II trajectories look almost.
 11. Figure S7a the similarities between WES and DEG is fairly limited to HLA's.
 12. Figure S7d both the axis and heat key are labeled as p value, is that correct?
 13. For the comparison between different types of achalasia, no n is mentioned from the groups.
 14. A number of pathway analysis are used throughout the paper such as GO and KEGG, that are different from figure to figure. This should be consistent.
 15. The names of clusters changes throughout the manuscript sometimes has numbers in it sometimes key genes, needs to be consistent.

Reviewer #2 (Remarks to the Author):

The manuscript by Liu et al. presents a single cell RNA-seq data set of patients with achalasia, a rare motility disorder of the oesophagus, which is caused by degenerated myenteric neurones. The authors sampled patients along disease progression and performed single cell RNA-sequencing to study the dynamic changes in the heterogeneity and to identify cell types associated with the disease after comparison to control samples from patients with benign leiomyomas. The authors identified C1QC macrophages and Tissue resident memory T cells to be significantly enriched in the oesophageal tissue of achalasia patients. C1QC macrophages were characterized by gene expression signatures that linked them to a neurodegenerative phenotype and thus provided an explanation for the degenerative characteristics of achalasia. Furthermore, the authors identify the C1QC macrophages to be most closely related, and based on pseudo-time trajectory analysis, to be derived from resident macrophages of the gut with functional similarities to microglia. Furthermore, the authors analysed tissue resident memory T cell and found that this subset of effector T cells expanded specifically in the diseased tissue.

The overall data set appears to be of high quality and the sampling design, in regards to the control tissue, is sound, with some additional questions remaining (see below). However, the authors draw their conclusions based on the scRNA-seq data only, thereby rendering the manuscript descriptive by nature, with no functional work-up. Several major and minor concerns prompt this reviewer to doubt that the authors would be able to add sufficient functional data addressing the role of C1QC

macrophages and effector TRMs in achalasia to a degree and in an acceptable time frame that it should be considered for publication in Nature Communications. If no changes were applied to the current manuscript following the points below this manuscript is more appropriate for a journal with specific clinical focus, as the discovered biomarkers might be of clinical relevance for the staging of the disease.

Major points:

- 1) The patient cohort explanation in the results section, methods section and Figure 1 caption needs more precision. The control tissue originates from leiomyomas, which are benign tumors of smooth muscle origin. What is known about the immune landscapes of this type of benign tumors in the literature? What are the immune cell characteristics in those control samples relative to non-diseased tissue or at least relative to blood immune cells that were also sampled? However, this reviewer is very skeptical that comparisons of blood immune cells with any immune cell isolated from tissue context will result into meaningful gene signatures.
- 2) The conclusion (in the Results section, page 7) about the presence of infiltrated immune cells around the ganglia is not reflected in the figures. What exactly did the authors have in mind? In which experimental group did they observe this phenomenon?
- 3) Regarding the myeloid cells sub-clustering – Many studies addressing the immune landscapes in the tumor microenvironment identify SPP1+ and FCN1+ macrophages along with C1QC+ macrophages. Did the researchers in this study observe the expression of SPP1 and FCN1 on any of the myeloid subclusters? If not – what could be the reason?
- 4) Page 9 – Researchers derive conclusion that the C1QC+ macrophages are significantly enriched in the LES tissue of achalasia patients, however there is no statistic data supporting this conclusion. What is the statistical significance of the difference in C1QC+ macrophages between control and achalasia LES?
- 5) Page 10 - „Collectively, we found gut-resident markers of CD4 and TIM4 highly expressed in the C1QC+ and CXCL3+ macrophages clusters and the directional flow originated from Mono1 to CXCL3+/C1QC+ macrophages in pseudotemporal trajectory. Both indicated C1QC+ macrophages were gut-resident macrophages”. Authors didn't show plots illustrating the CD4 and TIM4 expression on macrophages. Please provide the exact t-SNE plots with the identified CD4+TIM4+ macrophages. The conclusion that those macrophages originate from gut is based on the study done in mice (the [11] reference cited by Authors). In my opinion there is too little data presented here to draw this conclusion. I'm quite certain that the authors do not plan to include a mouse model in this study and perform lineage tracing experiments to support their claim of gut macrophages giving rise to the C1QC macrophages found in the LES of achalasia patients. It should be toned down and maybe discussed with caution.
- 6) CD4 is known to be expressed in myeloid cells as shown in many scRNAseq data sets despite the fact that it is a known marker of CD4+ T cells. However, this expression is not necessarily reflected on the protein level. If the Authors suggest the presence of CD4+ (and TIM4+) C1QC+ macrophages it should be shown on protein level.
- 7) The authors claim that the macrophages are functionally related to microglia and harbor neurodegenerative potential, thereby linking them to the degenerative phenotype of achalasia. This needs to be demonstrated in-vitro. Can macrophages isolated from achalasia patients directly kill neurons? The alternative explanation is that the macrophages express complement factors that bind to auto-antibodies targeting cell surface proteins on the neurons in the LES, or the macrophages act via presenting autoantigens to the effector TRMs identified by the authors, which in turn attack the neurons in the TRM. All those possibilities need to be tested, auto-antibodies, if present in the serum identified, TRMs exposed to C1QC+ macrophages tested for their killing ability targeting neurons of the LES.
- 8) Fig. 4g – How exactly were the Trms defined by the authors? Why were the markers CD4, CD8, CD69 and CD103 chosen for IHC staining? Are the Trm cells really mainly restricted to the myenteric plexus area? It is not obvious from the pictures presented and should be quantified by measuring, for example, the distance between Trms cells and PGP9.5+ cells. How did the Trm localization look like in control patient group? Is the myenteric plexus localization of Trm specific for the achalasia patients only?

9) More details on the pathophysiology of the disease with illustrations at which part of the esophagus achalasia mainly occurs would help to understand parts of the results and the discussion as it currently is written.

Minor Points:

1. The researchers identified one cluster of stem cells (Fig. 1b,c), however this a very general term. What type of stem cells was identified?
2. Page 8 - "Detailed cluster lineages in different tissue contexts are shown in Figure S1a." – I think the authors wanted to refer to Figure 2b.
3. Page 9 – "We found the functional homogeneity but more heterogeneity among myeloid cells in achalasia". – This reviewer is not sure what the authors intentions are here.
4. Fig. 3a –Marta Olah and David Gosselin refers to different microglia gene sets, this is not well described in the Figure caption.
5. Page 20 – "Previous studies indicated that type I might progress from type II and represent the end stage of achalasia⁴¹⁻⁴³." – what are the characteristics of the end stage of this disease?
6. In general, the manuscript requires additional proof reading of a native speaker.

Reviewer #3 (Remarks to the Author):

There is still not much known about mechanisms of development of achalasia, an esophageal motility disorder of unknown etiology with pathogenesis involving T- lymphocyte- mediated loss of ganglion cell in the myenteric plexus of the lower esophageal sphincter (LES). Despite the fact that T- lymphocytes, including CD8+ T-cells, have been shown to be involved in inflammation, the precise nature of inflammatory immune cells and their transcriptional profiles are not known. In this paper the authors describe specific immune cells with unique transcriptional phenotypes using single cell RNA sequencing. In particular they show unique infiltration around the ganglia by C1QC+ macrophages and by specific CD4+ and CD8+ tissue-resident memory T (Trm) cell clusters, such as ZNF683+ CD8+Trm and XCL1+ CD4+Trm. C1QC+ macrophages had high expression of brain microglial gene set and exhibited similar transcripts that are seen in dysfunctional microglia in Alzheimer disease and multiple sclerosis raising a possibility of a role in achalasia pathogenesis. The Trm cell clusters from patients with achalasia, particularly ZNF683+ CD8+ Trm and XCL1+ CD4+Trm, expressed transcripts of activation and proliferation that are typical of inflammatory phenotypes. In addition, these Trm clusters were clonally expanded. These findings are novel and allow a much more nuanced insight into the immune cell type and function in achalasia. The study will be of significance to the field providing essential data needed for furthering the investigation of the pathogenesis of achalasia.

Comments:

1. One general comment is about the use of the term "neurodegenerative disease" or "neurodegenerative" in relation to achalasia. It is not common in medical literature to define achalasia as a neurodegenerative disease. Ganglion loss is thought to be due to inflammation, which is believed by many to be autoimmune. There is no data that I know of describing an intrinsic abnormality of ganglion cells in achalasia to deserve the term 'neurodegenerative'. I think, it is better to adhere to the terminology which implies that a neurodegenerative disease is an intrinsic disease of neurons leading to their demise.
2. Another general comment is about conclusions that suggest a possible "important" or "crucial" role of macrophages or Trm clusters in achalasia. Without data on interactions between the immune cells and ganglion cells it is better to tone down some conclusions to something like: 'may have a role' or 'may be involved'.
3. Page 8, last paragraph: "Patients with benign leiomyomas originating from LES served as controls to ensure no invasion of the normal tissue". This is unclear. Do the authors mean that they used only normal LES tissue from the specimens with

leiomyoma? If so, how far was normal tissue from the leiomyoma? This is important for assessment of the validity of the control tissue.

4. Page 11. End of first paragraph: "... we speculate that the C1QC+ macrophage cluster was functionally similar to microglia and might have a crucial impact on the ENS."

Since enteric nervous system was just only once mentioned earlier in the paragraph, I do not think that there is enough data to speculate that the C1QC+ macrophage cluster "might have a crucial impact on the ENS".

5. Page 17. The start of the second paragraph. "Since the clonal expansion of leukocytes is evidence of pathogenesis,..."

I think this phrase sounds too broad and vague and is better not used.

6. Page 22. The end of the first paragraph: "Since this is the first study to show the importance of LES-infiltrated macrophages in the etiology of achalasia, ..."

It is more likely "pathogenesis" than "etiology".

Thanks for the reviewer's constructive and helpful advices. We have considered the reviewers' comments carefully and tried our best to revise the manuscript as the reviewers suggested.

First, substantial multicolor immunohistochemistry experiments and quantifications were added to validate the phenotypes of C1QC+ macrophages and Trms during the revision. To conduct a high-dimensional validation of the scRNA-seq findings, time-of-flight mass cytometry of immune cells in the LES tissue was added and analyzed. Moreover, we also performed a deeper bioinformatic analysis of the current scRNA-seq data as the reviewers suggested. Finally, other reviewer's concerns, such as organization and language of the manuscript, were comprehensively revised and modified.

Here are the detailed responses for the reviewers.

REVIEWER COMMENTS

Reviewer #1 (Remarks to the Author):

The manuscript by Liu et al profiles the LES and blood immune cell transcriptional landscape in achalasia using single-cell RNAseq. The authors identify both cellular and transcriptional features associated with achalasia, including increased frequencies of C1QC+ macrophages and inflammatory Trm cells, especially ZNF683+ CD8+ and XCL+ CD4 Trms. They further suggest that the C1QC+ achalasia enriched macrophage share gene expression profiles with microglia and have features similar to a number of neurodegenerative diseases including MS. The study also identifies clonal enrichment of T cells in cases with achalasia and proposes that type I achalasia progresses from type II by the nature of having a more inflammatory signature.

This study provides an excellent resource of single-cell RNAseq dataset of gene expression profiles and TCR/BCR profiles in blood and LES tissue of achalasia and control patients and identifies perhaps key immune features of the disease. However, there are a number issue including lack of validation of the key findings and over interpretation of the data that detract from the manuscript. Additionally, the organization of the manuscript (general characterizing of LES/blood immune cell and comparison between disease and control are back and forth) and grammatical errors make it challenging to interpret at times. There are also some missed opportunities with their datasets that should be explored further including more detailed TCR analysis and blood signature analysis. Finally, the figures would benefit from increased resolution and improved labeling of data.

Major points:

1. At a minimum, the two key findings of C1QC+ macrophages and ZNF683+ CD8+ and XCL1+ CD4 Trms need to be validated. The current IHC data in the paper is not specific to these cell types and not very convincing.

Response: Thanks for the reviewer's advice.

Gut-resident macrophages are resident within the gastrointestinal tract and have long been known to play a crucial role in the response to exogenous antigens. However, recent advances in scRNA-seq technology have revealed that resident macrophages throughout the gut are functional heterogeneity based on the niche they occupy¹. Shaw found that CD4 and TIM4 were canonical markers for gut-resident macrophages and are crucial for the differentiation and development of macrophages². Shaw et al. and Bujko et al. performed transcriptional signatures of gut-resident macrophages in human and mouse^{2,3}. Schepper et al. performed scRNA-seq and found one cluster of gut-resident macrophages, close to enteric neurons, could support enteric neurons and control essential intestinal functions of motility and secretion. They named such gut-resident macrophages as neuronal-associated gut-resident macrophages. They also performed single cell transcriptional signature of the neuronal-associated gut-resident macrophages⁴. Domanska et al. performed scRNA-seq and found neuronal-associated colonic macrophages were similar to gut-resident macrophages. In this study, they also revealed transcriptional signature of neuronal-associated colonic macrophages⁵.

In our revision, we first found C1QC+ macrophages expressed transcriptional signature similar to those of previously reported gut-resident macrophages in human and mouse, reported by Shaw and Bujko^{2,3} (Figures S3a and S3b). We also found C1QC+ macrophages expressed transcriptional signature similar to neuronal-associated gut-resident macrophages and neuronal-associated colonic macrophages reported by Schepper and Domanska^{4,5} (Figures S4a and S4b). In the studies of Schepper and Domanska, the neuronal-associated gut-resident macrophages and neuronal-associated colonic macrophages both highly expressed C1QC, indicating the same clusters with C1QC+ macrophages in our study⁴. We added the tSNE plots of CD4 and TIM4, canonical markers for gut-resident macrophages, showing the specific expressions of them in C1QC+ macrophages (Figure S3c). We also validated C1QC+ macrophages highly expressed CD4 and TIM4 at protein level by multicolor IHC of CD68, C1QC, CD4, and TIM4 (Figure S3d).

Collectively, we found C1QC+ macrophages expressed transcriptional signature similar to those of previously reported gut-resident macrophages, and canonical markers of CD4 and TIM4 (at both transcriptional and protein levels). Both indicated the C1QC+ macrophages might be gut-resident macrophages.

Among the marker genes of C1QC+ macrophages, C1QC, TMEM119, P2RY12, and TREM2 are the most accepted canonical markers of microglia⁶. TMEM119 is a type I transmembrane protein specifically expressed by resident microglia in the homeostatic condition, but not by blood-borne macrophages⁷. P2RY12 is a purinergic receptor that responds to ADP/ATP to increase cell migration, which is found to be exclusively expressed by microglia in the murine brain and is consistently expressed by human microglia throughout development^{8,9}. C1QC is another marker for microglia, which is important for the functions of developmental synapse pruning, disease-associated synapse loss and cognitive decline in ageing mediated by microglia⁶. TREM2 is specific expressed in microglia in the brain, which regulates microglial functions of sensing, housekeeping, and host defense, and dysregulation of TREM2 increases the risk of multiple neurodegenerative diseases¹⁰. We validated C1QC+ macrophages highly expressed C1QC, TMEM119, and P2RY12, by multicolor IHC in our revised manuscript (Figure 3b). Besides, we also validated the C1QC+ macrophages by high expression of C1QC and CD68 in tSNE plots of CyTOF. We also found C1QC+ macrophages highly expressed TREM2 and TMEM119 in CyTOF (Figure S1d). The multicolor IHC and CyTOF both validated the transcriptionally key cluster of C1QC+ macrophages.

We counted the expression of C1QC+ macrophages in multicolor IHC and confirmed the higher expression in the lower esophageal sphincter (LES) tissue of achalasia (Figure 2e). C1QC+ macrophages were mostly located surrounding the myenteric plexus of the LES in achalasia. Further CyTOF analysis showed the C1QC+ macrophages had a tendency of increased proportion in the LES of achalasia compared with controls (Figure S1d).

For the Trms, we validated the Trms by positive expressions of CD3, CD4, and CD69, or CD3, CD8, and CD69 in both multicolor IHC and CyTOF. Besides, we also validated the higher expression of Trms in the LES of achalasia compared with controls by histological quantifications of multicolor IHC (Figures 4g and 4h). The highly expressed Trms were mainly located surrounding the myenteric plexus of the LES in achalasia. Further CyTOF analysis showed Trms had a tendency of increased proportion in the LES of achalasia compared with controls (Figure S1d).

CD69 is the most accepted canonical marker to distinguish the Trms^{11,12}. In our previous submission, besides CD69, we also added CD103 in the panel, which was also reported as a canonical marker of Trms^{11,12}. However, although CD103 is important for the function of Trms, it was not expressed in all Trms both in our and previous scRNA-seq studies (Figure S5c)¹¹⁻¹³. Therefore, we deleted CD103 and only used CD69 to distinguish the Trms in our revised manuscript.

ZNF683 and XCL1 were transcriptionally most distinct markers and highly expressed in specific Trm clusters in our study. Previously, ZNF683 and XCL1 were also used for annotations of specific T cell clusters in scRNA-seq studies^{14,15}. Therefore, we used these two markers for annotations of specific Trms clusters. However, although with the transcriptionally distinct expressions in specific Trm clusters, we failed to achieve positive results of IHC staining and CyTOF for ZNF683 and XCL1. ZNF683 is a transcript factor in the nuclear and XCL1 is a secretory protein. We then searched the publications and found previous scRNA-seq studies did not validate ZNF683 and XCL1 at protein level¹⁴⁻²⁰, which might be caused by the difficulty of validation. Besides, we did not find any publications of the IHC staining of ZNF683 and XCL1, which indicated the difficulty of IHC staining of ZNF683 and XCL1. The phenomenon of discrepant transcriptional and protein levels also existed in previous studies²¹⁻²³. Previous studies found that transcript levels by themselves were not sufficient to predict protein levels in many scenarios, which might be caused by post-transcriptional modifications, protein's half-life, translation rates, protein synthesis delay, and protein transport^{22,23}.

a Human gut-resident macrophages signature (Bujko et al.)

b Mouse gut-resident macrophages signature (Shaw et al.)

c

d

a Neuronal-associated gut-resident macrophages signature (Schepper et al.)

b Neuronal-associated colonic macrophages signature (Domanska et al.)

2. The authors presume functionality from transcriptional data which is not directly possible. For example, the authors state the C1QC+ macrophages are functionally similar to microglia. Functional experiments

would need to be performed to make such conclusions. Currently, the cell types can only be claimed to have similar transcriptional signatures.

Response: Thanks for the reviewer's advice. In our current study, although we validated the similarity between C1QC+ macrophages and microglia by comparing transcriptional signatures and canonical markers of C1QC, TMEM119, P2RY12, and TREM2 by multicolor IHC and CyTOF, however, as the reviewer said, we did not perform functional experiment in our study, which is a limitation of our study. We added the limitation in the discussion as "... However, no functional experiment was performed to verify the functional similarity between C1QC+ macrophages and microglia in our study". Therefore, we took the reviewer's advice and changed the description of "functionally similar to microglia" as "transcriptionally similar to microglia".

3. The increase in TCR clonality is interesting but the data is only shallowly interpreted. What are the clones, are they public, private? Are there shared clones between patients, between tissue and blood. The manuscript would benefit from a much more in-depth TCR investigation.

Response: Thanks for the reviewer's constructive advice. Analysis of the TCR repertoire is one of our main points in the process of revision. We added 3 new parts of TCR analysis: clonal expansions but a decreased diversity of TCR repertoire in achalasia, patient-specific TCR repertoire, and a tendency of more human cytomegalovirus-specific CDR3 sequences in achalasia. Detailed findings are in the Results section of manuscript.

First, we found clonal expansions but a decreased diversity of TCR repertoire in achalasia. We found more clonal expanded TCR repertoire in tissue than in blood and more clonal expansions in achalasia than in controls (Figure 6a, Table S7). The Gini index, positively associated with TCR clonality²⁴, was a significantly higher in patients compared with that in controls (Figure 6b). Shannon's entropy index, positively correlated with TCR diversity²⁴, was significantly lower in achalasia than that in controls (Figure 6b). Collectively, these results indicated markedly clonal expansions but a decreased diversity of TCR repertoire of T cells in achalasia, in accordance with a previous study²⁵.

Second, we found patient-specific TCR repertoire in achalasia. We found dominant clones overlapped between tissue and blood for the same patient, but shared only at very low levels among different patients (Figure 6c). Different subclusters of T cells had shared clones (Figure 6d), indicating the same pseudotemporal cell trajectory. These results indicated clonal expansions of T cells was patient-specific, "private" to individual achalasia patients, in accordance with a previous study²⁵.

Thirdly, we found a tendency of more human cytomegalovirus (CMV)-specific CDR3 sequences in achalasia than those in controls by using VDJdb and TCRmatch methods (Figure 6e). Analysis of VDJdb methods showed the total virus-related sequences were more enriched in achalasia than those in controls, in expanded but not in non-expanded TCR repertoire (Table S7). The top four prevalent virus epitopes enriched in patients were KLGGALQAK (CMV), GILGFVFTL (influenza A virus), NLVPMVATV (CMV) and FRCPRRFCF (CMV) (Table S7). The findings of VDJdb method were confirmed by TCRmatch method (Figure 6e). These results indicated CMV antigen might be a potential driver of T cell clonal expansions in achalasia.

The most prevalent viruses in patients and controls

Antigen species	Non-expanded in case, n (%)	Expanded in case, n (%)	Sum in case, n (%)	Non-expanded in control, n (%)	Expanded in control, n (%)	Sum in control, n (%)
Total	17789	22366	40155	6882	6146	13028
CMV	204(1.15)	1867(8.35)	2071(5.16)	91(1.32)	68(1.11)	159(1.22)
Homo Sapiens*	51(0.29)	219(0.98)	270(0.67)	14(0.20)	44(0.72)	58(0.45)
Influenza A	43(0.24)	139(0.62)	182(0.45)	13(0.19)	5(0.08)	18(0.14)
SARS-CoV-2	52(0.29)	39(0.17)	91(0.23)	11(0.16)	15(0.24)	26(0.20)
EBV	57(0.32)	29(0.13)	86(0.21)	25(0.36)	23(0.37)	48(0.37)
HIV-1	24(0.13)	16(0.07)	40(0.10)	7(0.10)	0	7(0.05)
DENV1	4(0.02)	24(0.11)	28(0.07)	5(0.07)	0	5(0.04)
DENV3/4	3(0.02)	24(0.11)	27(0.07)	4(0.06)	3(0.05)	7(0.05)
HCV	14(0.08)	2(0.01)	16(0.04)	4(0.06)	0	4(0.03)
YFV	8(0.04)	0	8(0.02)	3(0.04)	2(0.03)	5(0.04)
MCMV	6(0.03)	0	6(0.01)	2(0.03)	0	2(0.02)
HIV	5(0.03)	0	5(0.01)	1(0.01)	0	1(0.01)
HTLV-1	3(0.02)	0	3(0.01)	0	0	0
DENV2	1(0.01)	0	1(0)	1(0.01)	3(0.05)	4(0.03)
M.tuberculosis	1(0.01)	0	1(0)	0	0	0
synthetic	1(0.01)	0	1(0)	0	0	0
LCMV	0	0	0	0	0	0
RSV	0	0	0	1(0.01)	0	1(0.01)

*Homo Sapiens (virus types as antigen species), with annotation "Patient species of the antigen, to the best clade resolution possible (e.g. HIV-1, HIV1*HXB2)", as stated at <https://github.com/antigenomics/vdjbdb/blob/master.README.md>.

The most prevalent virus epitopes in patients and controls

Epitopes	Antigen	Source organism	Clone size		P value (higher in case versus control)
			Case	Control	
KLGGALQAK	55 kDa immediate-early protein 1	Human cytomegalovirus (CMV)	1846	121	1.34E-106
GILGFVFTL	Matrix protein 1	Influenza A virus	129	11	5.47E-07
NLVPMVATV	HCMVUL83	Human cytomegalovirus (CMV)	209	27	6.37E-07
FRCPRRFCF	UL28	Human cytomegalovirus (CMV)	41	0	1.34E-05

4. The blood data is very superficially analyzed throughout the manuscript. Having a biomarker for achalasia and particularly more severe achalasia would potentially be of benefit. However, little is mentioned about the peripheral blood signatures unique to achalasia throughout the manuscript. A more in-depth analysis would be of benefit here.

Response: Thanks for the reviewer's advice. Previously, because it is difficult to acquire the specimen of the LES tissue of achalasia, previous studies about the immune cells of achalasia in the LES were extremely limited. Besides, the cell proportions and transcriptome of the LES tissue were very different between achalasia and controls in our study. Therefore, we mainly focused on the LES tissue analysis in our previous analysis.

We took the reviewer's advice and added the blood analysis in the revised manuscript, including the different transcriptome between peripheral blood and LES tissue, and achalasia-associated transcriptional changes of immune cells between achalasia and controls in peripheral blood. Detailed descriptions are as follows:

Immune cells in peripheral blood showed different transcriptome with that in LES tissue. Whereas, the differential expressed genes between tissue and blood were quite similar among different cell types (Figure S11a and Table S12). Heat shock proteins, tissue-specific markers of CD69, and pathways of IL-17, MAPK, and VEGFA-VEGFR2 were up-regulated in LES tissue, while WNT pathway was up-regulated in blood (Figures S11a and S11b).

We explored achalasia-associated transcriptional changes of myeloid cells between achalasia and controls in peripheral blood even though with comparable compositions (Figure 2b). Analysis of DEGs revealed the up-regulations of PDPF, NENF, and CYBA and down-regulations of FOS, DUSP1, and GPX1 in peripheral blood myeloid cells of achalasia (Figure S12a and Table S4). Metascape analysis showed increased expressed DEGs were most enriched in response of EIF2AK4 (GCN2) to amino acid deficiency, cellular response to cytokine stimulus, and type II interferon signaling (Figure S12a). Next, achalasia-associated transcriptional changes of lymphocytes in peripheral blood were analyzed, especially for GNLY+ CD8+ T cells, CD27+ B cells, and plasma, which were proportionally increased in achalasia (Figure 4d). Analysis of DEGs revealed the common up-regulations of SFPQ, KLF2, and CYBA and common down-regulations of JUNB, FAM118A, and C21orf33 in peripheral blood lymphocytes of achalasia (Figure S12b and Table S6). The most specific DEGs of GNLY+ CD8+ T cells were up-regulations of KLF2 and PDPF and down-regulations of KIR2DL3 and HLA-DRB5. CYBA and KLF2 were the common most up-regulated DEGs of CD27+ B cells and plasma, while the most down-regulated DEGs were RPS29 and JUNB for CD27+ B cells and IGKV5-2 for plasma (Figure S12c). The most enriched pathways of GNLY+ CD8+ T cells, CD27+ B cells, and plasma were adaptive immune response, cellular response to cytokine stimulus, and rRNA processing in the nucleus and cytosol (Figure S12c).

5. The manuscript would benefit from a thematic re-organization, improved resolution and labeling of figures and correction of grammatical errors throughout.

Response: Thanks. We improved the quality of the figures and revised the organization. Besides, the language was also polished by a native speaker.

Minor points:

1. A number of statements are made about data throughout the manuscript that are not shown in the figures. All data needs to be shown in the figures. Some examples:

Response: Thanks for the reviewer’s careful check. We apologized for the mistake. We double-checked the manuscript and added the data in the figures.

a. In the Single-cell transcriptomics of subclustered myeloid cells section: “We found expression of CD4 and TIM4 in the C1QC+ and CXCL3+ macrophages cluster”, however there are no figures showing CD4 and TIM4 macrophage expression.

Response: Thanks. The tSNE plots of CD4 and TIM4 were added in the revised figure.

b. A number of genes that the authors refer to are missing from the figures throughout the manuscript.

Response: Thanks. We double-checked the manuscript and added the data of genes in the figures.

2. The IHC data presented in the manuscript is hard to interpret as it shows only small areas with unconvincing staining. For example:

a. Fig 1f-g authors state that macrophages (CD68) were increased in achalasia. However, only 1 cell is positive in both the control and achalasia image that is shown. The ganglia staining (PGP9.5) is over-exposed in controls and hard to see in the case that is shown. So tricky to interpret their data to be infiltration of immune cells into the ganglion. From the image shown only the increase in CD4/8 in the achalasia image is convincing. However, neither CD8 or CD4 staining by itself can be used for definitive T cell identification as both markers are expressed in other cell types. CD4 for examples is expressed on intestinal macrophages (as the authors point out) and CD8 is expressed in NK cells.

Response: Thanks for the reviewer's advice. We apologized the previous figure was unclear.

We took the reviewer's advice and performed substantial multicolor IHC experiments and quantifications of C1QC macrophages and Trms during the revision.

For C1QC+ macrophages, we first found C1QC+ macrophages expressed gene set similar to those of previously reported gut-resident macrophages in human and mouse (Figures S3a and S3b)^{2,3}. Besides, we found C1QC+ macrophages transcriptionally highly expressed CD4 and TIM4 and validated at protein level by multicolor IHC, which are canonical markers for gut-resident macrophages² (Figures S3c and S3d). This finding together with the transcriptional signature of C1QC+ macrophages indicated C1QC+ macrophages might be gut-resident macrophages.

Next, we validated C1QC+ macrophages highly expressed C1QC, TMEM119, and P2RY12 by multicolor IHC, which are the most canonical markers of microglia⁶ (Figure 3c). This finding together with CyTOF and the transcriptional signature of C1QC+ macrophages indicated the phenotypic similarity between C1QC+ macrophages and microglia.

We counted the expression of C1QC+ macrophages in multicolor IHC and confirmed the higher expression in the LES tissue of achalasia (Figure 2e). Besides, this C1QC+ macrophages were mostly located surrounding the myenteric plexus of LES in achalasia.

For the Trms, we validated the Trms by positive expressions of CD3, CD4, and CD69, or CD3, CD8, and CD69 in the multicolor IHC and CyTOF. Besides, we also validated the higher expression of Trms in the LES of achalasia compared with controls by histological quantifications of multicolor IHC (Figures 4g and 4h). The highly expressed Trms were mainly located surrounding the myenteric plexus of the LES in achalasia.

CD69 is the most accepted canonical marker to distinguish the Trms^{11,12}. In our previous submission, besides CD69, we also added CD103 in the panel, which was also reported as a canonical marker of Trms. However, although CD103 is important for the function of Trms, it was not expressed in all Trms both in our and previous scRNA-seq studies (Figure S5c)¹¹⁻¹³. Therefore, we deleted CD103 and only used CD69 to distinguish the Trms in our revised submission.

Loss of myenteric neurons and immune cells infiltration in the LES is the main pathophysiology in achalasia. In achalasia, these myenteric neurons are decreased in number or even absent^{26,27}. Although its cause remains largely unknown, ganglionitis resulting from an aberrant immune response has been proposed to underlie the loss of LES myenteric neurons^{26,27}. Therefore, we took image of residual sparse neurons and nerve fibers in our previous figure, which was easily seen in achalasia. We also took the

reviewer's advice and used clearer image of the neurons or nerve fibers in the new figures. We updated the figure in the revised manuscript.

b. Fig 3b only shows images of an achalasia case without any control images and would need to show the comparison to validate increase in C1qc macrophages in achalasia. Trem2 and C1qc staining is remarkably similar suggesting that one is perhaps the background of the other.

Response: Thanks. We apologized the previous figure was unclear. We added the control images and made quantifications in the new figure.

In our revised manuscript, we validated C1QC+ macrophages highly expressed C1QC, TMEM119, and P2RY12 by multicolor IHC (Figure 3c), which are the most canonical markers of microglia⁶ and could help to show the similarity of C1QC+ macrophages and microglia.

We counted the expression of C1QC+ macrophages in multicolor IHC and confirmed the higher expression in the LES tissue of achalasia (Figure 2e). Besides, this C1QC+ macrophages were mostly located surrounding or possibly infiltrating the myenteric plexus of LES in achalasia.

c. Presence of the macrophages near the ganglion cells does not by themselves suggest that the pathophysiology of the disease is neurodegeneration.

Response: Loss of myenteric neurons and immune cell infiltration in the LES is the main pathophysiology in achalasia. Although its cause remains largely unknown, ganglionitis resulting from an aberrant immune response has been proposed to underlie the loss of LES myenteric neurons^{26,27}.

We found many immune cells, including macrophages and T cells, near the residual ganglia or sparse neurons and nerve fibers, which was in accordance with the pathophysiology of neuroinflammation. However, we admitted the phenomenon of macrophages near the ganglia by itself could not suggest that the pathophysiology of the disease is neurodegeneration. Since this description was inappropriate, we deleted this sentence in the revision.

d. Figure 4g. CD103 is only on 1 cell. CD8/CD4 staining even in combination with CD69 (which is not on all the cells) does not confirm that these cells are TRMs. Again, no data on control vs achalasia here.

Response: Thanks.

For the Trms, we validated the Trms by positive expressions of CD3, CD4, and CD69, or CD3, CD8, and CD69 by multicolor IHC. We also validated the higher expression of Trms in the LES of achalasia compared with controls by histological quantifications of multicolor IHC (Figures 4g and 4h). The highly expressed Trms were mainly located surrounding the myenteric plexus of the LES in achalasia.

CD69 is the most accepted canonical marker to distinguish the Trms^{11,12}. In our previous submission, besides CD69, we also added CD103 in the panel, which was also reported as a canonical marker of Trms. However, although CD103 is important for the function of Trms, it was not expressed in all Trms both in our and previous scRNA-seq studies (Figure S5c)¹¹⁻¹³. Therefore, as the reviewer suggested, we deleted CD103 and only used CD69 to distinguish the Trms in our revised submission.

3. There are numerous sentences throughout the manuscript that are challenging to interpret and should be restated. For example:

a. “We found the functional homogeneity but more heterogeneity among myeloid cells in achalasia”?
Hard to understand as they’re only showing the transcriptional profiles, no functional assay.

Response: We were so sorry for the unclear description. We deleted this sentence in the revised manuscript. We also doubled-checked and revised the organization and the language was polished by a native speaker.

4. Figure 3 needs to be quantified.

Response: Thanks. We added the quantifications of C1QC+ macrophages expressions of multicolor IHC in the achalasia and controls. We found C1QC+ macrophages were more highly expressed in the LES tissue of achalasia compared with controls (Figure 2e).

5. No graph in main Figure 4 showing the maker genes used to identify different T cell types.

Response: Based on marker gene expression, the 16 T cell clusters were further categorized into 6 main cell types, including naive T cells (CCR7), central memory T cells (CD27, CD44, CCR7+), effector memory T cells (CD27, CD44, CCR7-), T regulator cells (Tregs) (FOXP3), Trms (CD69), and effector T cells (GNLY). These markers genes of 6 main cell types were show below. All the marker genes of the 16 T cell clusters were shown in Figures 4c and 4d.

To make it easily read and found, we emphasized the maker genes in the manuscript. “Based on marker gene expression, the CD4+ and CD8+ T cell clusters were further categorized into 6 main cell types, including naive T cells (CCR7), central memory T cells (CD27, CD44, CCR7+), effector memory T cells (CD27, CD44, CCR7-), T regulator cells (Tregs) (FOXP3), Trms (CD69), and effector T cells (GNLY). (Figures 4c and 4d, and Tables S2 and S6)”

6. Figure 4e is poorly described. From the pseudotime trajectory, most Trms are in transitional stages in naïve cell differentiating to effector memory T cells pathway. Tc7 cells were branch out, suggesting the unique features (cycling genes) compared to other T cells.

Response: Thanks for the reviewer's advice.

The pseudotime trajectories of T cells were divided into all T cells and Trms in our study. In the old Figure 4e (new Figure 4f), we mainly described the trajectory of all T cells. The pseudotemporal trajectory of T cells showed a strong directional flow from the naive to the memory and finally to the effector state (new Figure 4f), as seen previously²⁸. As the reviewer said, the pseudotemporal trajectory of Tc7 cells and other Trms were detailed described in the section of Trms in the new Figures S5f and S5g.

That is, “Analysis of the pseudotemporal transcriptional trajectories of CD4+ and CD8+ T cells showed that most Trm subclusters originated from effector memory T cells, while TYMS+ CD8+ Trms branched out in pseudotime, indicating the specifically proliferated state (Figures S5f and S5g). Heat maps also showed continuous phenotypic variation along pseudotime and the cell type-specific changes (Figure S5g). The naive and central memory T cell markers such as CCR7 and SELL were expressed early, while expression of activation markers such as GNLY, increased along pseudotime. A series of genes were highly expressed in Trms, including XCL1, FOS, FOSB, TSC22D3, JUN, NR4A2, suggesting their potential functions for Trms.”

7. DEGs in Figure 5 are these for Trm in achalasia VS Trm in controls or Trm VS other T cells in achalasia?

Response: Sorry for the unclear description. The term of DEGs in the manuscript all represented differentially expressed genes between achalasia and controls. We deleted the old Figure 5b as it was not necessary and hard to understand. In the new Figure 5, Figure 5a depicted DEGs for different lymphocyte clusters, showing that Trms had more DEGs than other lymphocytes; 5b depicted dot plot of DEGs for the 26 lymphocyte clusters in blood and LES tissue; 5c depicted volcano plot showing the merged DEGs for Trms; 5d depicted specific DEGs for Trms, GZMK+ CD8+ T cells (Tc2), and GNLY+ CD8+ T cells

(Tc8); 5e depicted GO (up) and KEGG (down) analysis of DEGs for the 26 lymphocyte clusters; 5f depicted Metascape analysis of DEGs for Trms.

We revised the description in the manuscript and figure legend, and modified the labels in the figure to be clearer and more easily understood. In the figure legend of Figure 5, “a, A bar chart depicting the DEGs between achalasia and controls for the 26 lymphocyte clusters. b, Dot plot of DEGs for the 26 lymphocyte clusters in blood and LES tissue. c, Volcano plot showing the merged DEGs for Trms. d, Specific DEGs for Trms, GZMK+ CD8+ T cells (Tc2), and GNLY+ CD8+ T cells (Tc8). e, GO (up) and KEGG (down) analysis of DEGs for the 26 lymphocyte clusters. f, Metascape analysis of DEGs for Trms.”.

8. Figure 5. f only one pathway analysis is shown is this KEGG? GO? Unclear and both are stated in the text. g,e the conclusion that these cells are pro-inflammatory is not supported fully by the data as a number of inflammatory cytokines were not upregulated in these clusters.

Response: Old Figure 5f (new Figure 5e) had a thin white line in the middle, which separated the GO (up) and KEGG (low) results. We are sorry for the unclear label. We made it more clear in the revised figure as follows. As the reviewer said, the Figures 5d-f showed some pro-inflammatory pathways and cytokines were upregulated, while some others were not upregulated or even downregulated. These results indicated the Trms presented dysregulated immune responses in achalasia, instead of only a pro-inflammatory state. We changed the related descriptions in the abstract, results, and discussion. Thanks for the reviewer's constructive advice.

9. Figure 6 e. the sentence that clonal expansion is evidence of pathogenesis is not correct. One can have expanded T cells clones that do not cause disease.

Response: Thanks for the reviewer's reminder. We deleted this sentence in the revision.

10. Unclear how to interpret data in Figure 7f as type I and type II trajectories look almost.

Response: Sorry for the unclear description and label. Previous studies indicated that type I might progress from type II and represent the late stage of achalasia. Therefore, we drew the pseudotemporal trajectories of each cell type based on different types of achalasia.

In the previous figure as below, the X-axis represented different cell types. The Y-axis represented pseudotime of each cell type. The higher location of the Y-axis indicated later pseudotime. For each cell type, the pseudotime was divided by type I and II achalasia. The pseudotime was earlier in type II than type I achalasia in most clusters, no matter myeloid cells or lymphocytes. This result was in accordance with previous studies, which indicated that type I might progress from type II.

We changed the Figure 7f as below and the related description to make it easier understood.

Old Figure 7f

New Figure 7f

11. Figure S7 the similarities between WES and DEG is fairly limited to HLA's.

were illustrated. b, c, Specific DEGs for C1QC+ macrophages (b) and Trms (c) overlapped with risk genes for achalasia identified by WES. d, Significant levels of the overlap between DEGs of achalasia and the risk genes for other neurodegenerative diseases from the GWAS catalog (<https://www.ebi.ac.uk/gwas/>), including AD, PD, ALS, and MS. Both the axis and heat key are represented as p value (Fisher's exact test)."

13. For the comparison between different types of achalasia, no n is mentioned from the groups.

Response: We added the number of different types of achalasia in the Results, Methods and Figure 7 legend.

14. A number of pathway analysis are used throughout the paper such as GO and KEGG, that are different from figure to figure. This should be consistent.

Response: Thanks. We used different pathway analyses to verify each other and used specific pathway analyses when necessary. The new Figures 2g, 2h, and 5e used KEGG analysis, and only new Figure 5e used GO analysis. Other pathways were all analyzed by Metascape analysis. For different pathway analyses, we labeled it clearer in the figures and described more detailed in the manuscript to make it easily understood.

15. The names of clusters change throughout the manuscript sometimes has numbers in it sometimes key genes, needs to be consistent.

Response: Thanks. We revised the names of clusters to be consistent. All clusters were named using marker genes.

Reviewer #2 (Remarks to the Author):

The manuscript by Liu et al. presents a single cell RNA-seq data set of patients with achalasia, a rare motility disorder of the oesophagus, which is caused by degenerated myenteric neurons. The authors sampled patients along disease progression and performed single cell RNA-sequencing to study the dynamic changes in the heterogeneity and to identify cell types associated with the disease after comparison to control samples from patients with benign leiomyomas. The authors identified C1QC macrophages and Tissue resident memory T cells to be significantly enriched in the oesophageal tissue of achalasia patients. C1QC macrophages were characterized by gene expression signatures that linked them to a neurodegenerative phenotype and thus provided an explanation for the degenerative characteristics of achalasia. Furthermore, the authors identify the C1QC macrophages to be most closely related, and based on pseudo-time trajectory analysis, to be derived from resident macrophages of the gut with functional similarities to microglia. Furthermore, the authors analysed tissue resident memory T cell and found that this subset of effector T cells expanded specifically in the diseased tissue.

The overall data set appears to be of high quality and the sampling design, in regards to the control tissue,

is sound, with some additional questions remaining (see below). However, the authors draw their conclusions based on the scRNA-seq data only, thereby rendering the manuscript descriptive by nature, with no functional work-up. Several major and minor concerns prompt this reviewer to doubt that the authors would be able to add sufficient functional data addressing the role of C1QC macrophages and effector TRMs in achalasia to a degree and in an acceptable time frame that it should be considered for publication in Nature Communications. If no changes were applied to the current manuscript following the points below this manuscript is more appropriate for a journal with specific clinical focus, as the discovered biomarkers might be of clinical relevance for the staging of the disease.

Major points:

1) The patient cohort explanation in the results section, methods section, and Figure 1 caption needs more precision. The control tissue originates from leiomyomas, which are benign tumors of smooth muscle origin. What is known about the immune landscapes of this type of benign tumors in the literature? What are the immune cell characteristics in those control samples relative to non-diseased tissue or at least relative to blood immune cells that were also sampled? However, this reviewer is very skeptical that comparisons of blood immune cells with any immune cell isolated from tissue context will result into meaningful gene signatures.

Response: Thanks for the reviewer's advice. Esophageal leiomyomas are benign tumors of smooth muscle with biologically localized growth and no invasion of surrounding normal tissues. We took the surrounding normal tissues of esophageal leiomyomas as controls in our study. Besides, to reduce the impact of leiomyomas at maximum, we took the surrounding normal tissues 5 mm away from the leiomyoma. Therefore, the surrounding normal tissues could consider normal tissues without being influenced by the leiomyomas.

By far, no immune landscape of surrounding normal tissues of leiomyomas was reported in the literature. In our recent publication, we compared the achalasia and controls and found CD8+ T cells infiltrated around residual ganglia in the LES in achalasia³¹. In this study, control LES tissues were also obtained from surrounding normal tissues of leiomyomas from patients who underwent endoscopic resection. We found the CD8+ T cells were very low infiltrated and the ganglia were normal in these control specimens³¹.

As the reviewer said, we admit non-disease LES tissue is an ideal specimen for controls. However, it is ethically impossible to take LES muscle tissue from non-disease-healthy people. Because the LES muscle tissue is covered by normal mucosa, the endoscopic biopsy cannot reach the muscle tissue. Therefore, the ideal non-disease LES tissue is very hard to acquire. Therefore, we took normal tissues from patients with leiomyomas, which are benign tumors originating from smooth muscle layer with biologically localized growth and no invasion of surrounding normal tissues. Besides, to reduce the impact of leiomyomas at maximum, we took the surrounding normal tissue 5 mm away from the leiomyoma. The surrounding normal tissues could consider normal tissues without influenced by leiomyomas.

As the reviewer said, comparisons of blood immune cells with any immune cells isolated from tissue context will result in meaningful gene signatures. We agreed with the reviewer's opinion. Because immune cells in blood had relatively different functions and immune microenvironment compared with

those in tissue, the immune cells in the blood were different from LES tissue, both in compositions and transcriptional signatures (Figures 1f and S11a). However, both the immune cells in blood and LES tissue from controls were considered normal and we mainly focused on the comparisons between achalasia and controls and explored the achalasia-specific changes.

Previously we mainly demonstrated the achalasia-specific changes in LES tissue, while during the revision, we further explored the achalasia-specific changes in blood. For myeloid cells, we found myeloid cells in blood were enriched to comparable composition in achalasia and controls (Figure 2b). Analysis of DEGs revealed the up-regulations of PDPF, NENF, and CYBA and down-regulations of FOS, DUSP1, and GPX1 in peripheral blood myeloid cells of achalasia (Figure S12a and Table S4). Metascape analysis showed increased expressed DEGs were most enriched in response of EIF2AK4 (GCN2) to amino acid deficiency, cellular response to cytokine stimulus, and type II interferon signaling (Figure S12a). For lymphocytes, we found the lymphocytes in blood were similar in achalasia and controls except the increased population of GNLY+ CD8+ T cells, CD27+ B cells, and plasma in achalasia (Figure 4d). Analysis of DEGs revealed the common up-regulations of SFPQ, KLF2, and CYBA and common down-regulations of JUNB, FAM118A, and C21orf33 in peripheral blood lymphocytes of achalasia (Figure S12b and Table S6). The most enriched pathways of GNLY+ CD8+ T cells, CD27+ B cells, and plasma were adaptive immune response, cellular response to cytokine stimulus, and rRNA processing in the nucleus and cytosol (Figure S12c).

We also took the reviewer's advice and explained the control cohort more precisely in the sections of Results, Methods and Figure 1 caption. We also modified the Figure 1a to be more precise.

2) The conclusion (in the Results section, page 7) about the presence of infiltrated immune cells around the ganglia is not reflected in the figures. What exactly did the authors have in mind? In which experimental group did they observe this phenomenon?

Response: We are sorry for the previous figure was unclear. We updated the figure in the revised manuscript. Please see the newly uploaded figure.

It is widely reported that loss of myenteric neurons and immune cell infiltration in the LES is the main pathophysiology in achalasia^{26,27}. In achalasia, these myenteric neurons are decreased in number or even absent. Mostly only residual sparse neurons or nerve fibers could be seen in the LES in achalasia. Although its cause remains largely unknown, ganglionitis resulting from an aberrant immune response has been proposed to underlie the loss of LES myenteric neurons^{26,27}. Therefore, we took images of residual sparse neurons and nerve fibers (shown by PGP9.5) in our previous figure, which is easily seen in achalasia. In our previous study, we also found CD8+ T cells infiltrated around residual ganglia in the LES in achalasia³¹, in accordance with our current study.

In our study, we counted the expression of C1QC+ macrophages in multicolor IHC and confirmed the higher expression in the LES tissue of achalasia (Figure 2e). Besides, this C1QC+ macrophages were mostly located surrounding the residual sparse myenteric plexus or nerve fibers of LES in achalasia. For the Trms, we validated the higher expression of Trms in the LES of achalasia compared with controls by histological quantifications of multicolor IHC (Figures 4g and 4h). The highly expressed Trms were mainly located surrounding the residual sparse myenteric plexus or nerve fibers of the LES in achalasia.

3) Regarding the myeloid cells sub-clustering – Many studies addressing the immune landscapes in the tumor microenvironment identify SPP1+ and FCN1+ macrophages along with C1QC+ macrophages. Did the researchers in this study observe the expression of SPP1 and FCN1 on any of the myeloid subclusters? If not – what could be the reason?

Response: Thanks. Zhang Zemin et al. reported SPP1+ tumor-associated macrophages (TAMs) associated with tumor angiogenesis in 8 tumor types, including breast cancer, pancreatic adenocarcinoma, and colon cancer¹⁴. In our study, the SPP1+ macrophages were very low (shown below). Maybe because the angiogenesis is normal and under control in the non-tumor environment, this cluster exists but does

not expand, and it may only expand in the tumor environment.

Compared with SPP1+ macrophages, the C1QC + TAMs exhibited significantly higher “phagocytosis scores” in previous study¹⁴, which is similar to our finding of C1QC+ macrophages. C1QC+ macrophages exhibited higher phagocytosis-related pathways compared with other myeloid cells in our study (Figure 3b).

Myeloid-derived suppressor cells (MDSCs) are pathologically activated neutrophils and monocytes with potent immunosuppressive activity. They are implicated in the regulation of immune responses in many pathological conditions and are closely associated with poor clinical outcomes in cancer³². Previously, MDSCs were reported with high expressions of S100A family genes, such as S100A8, S100A9, FCN1 and VCAN, and relatively low expressions of HLAs^{33,34}. We found all the monocytes and macrophages, except C1QC+ and CXCL3+ macrophages, highly expressed S100A family genes of S100A8, S100A9, FCN1 and VCAN, and relatively low expressed HLAs (Figures 2d and S2d), which indicated all the monocytes and macrophages, except C1QC+ and CXCL3+ macrophages, were MDSC-like monocytes/macrophages. Thanks for the reviewer’s constructive suggestion. We added the descriptions and figures in the manuscript.

4) Page 9 – Researchers derive conclusion that the C1QC+ macrophages are significantly enriched in the LES tissue of achalasia patients, however there is no statistic data supporting this conclusion. What is the statistical significance of the difference in C1QC+ macrophages between control and achalasia LES?
Response: Thanks for the reviewer’s suggestion. We added the quantification of C1QC+ macrophages in the LES tissue between controls and achalasia. We confirmed the higher expression of C1QC+ macrophages by quantification of multicolor IHC in the LES tissue of achalasia (Figure 2e).

5) Page 10 - ,, Collectively, we found gut-resident markers of CD4 and TIM4 highly expressed in the C1QC+ and CXCL3+ macrophages clusters and the directional flow originated from Mono1 to CXCL3+/C1QC+ macrophages in pseudotemporal trajectory. Both indicated C1QC+ macrophages were gut-resident macrophages”. Authors didn’t show plots illustrating the CD4 and TIM4 expression on macrophages. Please provide the exact t-SNE plots with the identified CD4+TIM4+ macrophages. The conclusion that those macrophages originate from gut is based on the study done in mice (the [11] reference cited by Authors). In my opinion there is too little data presented here to draw this conclusion. I’m quite certain that the authors do not plan to include a mouse model in this study and perform lineage tracing experiments to support their claim of gut macrophages giving rise to the C1QC macrophages found in the LES of achalasia patients. It should be toned down and maybe discussed with caution.

Response:

Gut-resident macrophages are resident within the gastrointestinal tract and have long been known to play a crucial role in the response to exogenous antigens. However, recent advances in single-cell RNA sequencing technology have revealed that resident macrophages throughout the gut are functional heterogeneity based on the niche they occupy¹. Shaw found that CD4 and TIM4 were canonical markers for gut-resident macrophages and are crucial for the differentiation and development of macrophages². Shaw et al. and Bujko et al. performed transcriptional signatures of gut-resident macrophages in human and mouse^{2,3}. Schepper et al. performed scRNA-seq and found one cluster of gut-resident macrophages, close to enteric neurons, could support enteric neurons and control essential intestinal functions of motility and secretion. They named such gut-resident macrophages as neuronal-associated gut-resident macrophages. They also performed transcriptional signature of the neuronal-associated gut-resident macrophages by scRNA-seq⁴. Domanska et al. performed scRNA-seq and found neuronal-associated colonic macrophages were similar to gut-resident macrophages. In this study, they also revealed transcriptional signature of neuronal-associated colonic macrophages⁵.

In our revision, we first found C1QC+ macrophages expressed transcriptional signature similar to those of previously reported gut-resident macrophages in human and mouse, reported by Shaw and Bujko (Figures S3a and S3b). We also found C1QC+ macrophages expressed transcriptional signature similar to neuronal-associated gut-resident macrophages and neuronal-associated colonic macrophages reported by Schepper and Domanska^{4,5} (Figures S4a and S4b). In the study of Schepper and Domanska, neuronal-associated gut-resident macrophages and neuronal-associated colonic macrophages both highly expressed C1QC, indicating the similar cluster with C1QC+ macrophages in our study.

We added the tSNE plots of CD4 and TIM4, canonical markers for gut-resident macrophages, showing the specific expressions of them in C1QC+ macrophages (Figure S3c). We also validated C1QC+ macrophages highly expressed CD4 and TIM4 at protein level by multicolor IHC of CD68, C1QC, CD4, and TIM4 (Figure S3d).

Collectively, we found C1QC+ macrophages expressed transcriptional signature similar to those of previously reported gut-resident macrophages, and canonical markers of CD4 and TIM4 (at both transcriptional and protein levels). Both indicated the C1QC+ macrophages might be gut-resident macrophages.

Thanks for the reviewer's advice, we toned down our description in the manuscript as "Both indicated the C1QC+ macrophages might be gut-resident macrophages."

a Human gut-resident macrophages signature (Bujko et al.)

b Mouse gut-resident macrophages signature (Shaw et al.)

c

d

6) CD4 is known to be expressed in myeloid cells as shown in many scRNAseq data sets despite the fact that it is a known marker of CD4+ T cells. However, this expression is not necessarily reflected at protein level. If the Authors suggest the presence of CD4+ (and TIM4+) C1QC+ macrophages it should be shown at protein level.

Response: Thanks. We validated C1QC+ macrophages highly expressed CD4 and TIM4 at protein level by multicolor IHC of CD68, C1QC, CD4 and TIM4 (Figure S3d).

7) The authors claim that the macrophages are functionally related to microglia and harbor neurodegenerative potential, thereby linking them to the degenerative phenotype of achalasia. This needs to be demonstrated in-vitro. Can macrophages isolated from achalasia patients directly kill neurons? The alternative explanation is that the macrophages express complement factors that bind to auto-antibodies targeting cell surface proteins on the neurons in the LES, or the macrophages act via presenting autoantigens to the effector TRMs identified by the authors, which in turn attack the neurons in the TRM. All those possibilities need to be tested, auto-antibodies, if present in the serum identified, TRMs exposed to C1QC+ macrophages tested for their killing ability targeting neurons of the LES.

Response: Thanks for the reviewer's advice. In our study, we first found that C1QC+ macrophages highly expressed previously identified microglial gene sets, including C1QA, C1QB, C1QC, TREM2, TMEM119, P2RY12, and GPR34 (Figure 3a). Multicolor IHC and CyTOF of the LES tissue confirmed this C1QC+ macrophage cluster highly expressed canonical markers of microglia of C1QC, TMEM119

and P2RY12, and TREM2. All these results indicated the phenotypic similarity between C1QC+ macrophages and microglia.

We counted the expression of C1QC+ macrophages in multicolor IHC and confirmed the higher expression in the lower esophageal sphincter (LES) tissue of achalasia (Figure 2e). Besides, this C1QC+ macrophages were mostly located surrounding or possibly infiltrating the myenteric plexus of LES in achalasia. These results indicated a potential impact of C1QC+ macrophages on the enteric nervous system.

Metascape analysis of DEGs showed that C1QC+ macrophages had increased expression of transcripts for TYROBY causal network in microglia, synapse pruning, phagosome, lysosome, and microglial pathogen phagocytosis pathway and microglial cell activation in achalasia (Figure 3d). These pathways highlighted C1QC+ macrophages in achalasia exhibited an activated state similar with microglia. The activated microglia-mediated synaptic pruning and loss is highly up-regulated in many neurodegenerative disorders^{10,35,36}, suggesting that a similar mechanism of complement-mediated synapse elimination might drive achalasia progression.

Dysregulation of microglia contributes to the pathogenesis of several neurodegenerative diseases, including AD, PD, and MS³⁷⁻³⁹. Given the similarities of C1QC+ macrophages and microglia, we compared transcripts of C1QC+ macrophages in achalasia with those of disease-associated microglia from other neurodegenerative diseases³⁷⁻⁴³. GSVA analysis showed that the signatures of DEGs for C1QC+ macrophages were similar to those of neurodegenerative dysfunctional microglia (Figure 3e and Table S5). Many disease-associated genes in microglia were differentially expressed in C1QC+ macrophages in achalasia (Figure 3f). Among these, multiple M0-homeostatic microglial transcripts were repressed in C1QC+ macrophages, including CST3, NFKB1, JUN, and EGR1 (Figure 3d), while the key transcripts of dysfunctional microglia in neurodegenerative diseases, including APOE, TREM2, and LGALS3, were strongly up-regulated⁴³ (Figure 3d). Among these, *EGR1* was the master transcription factor in M0-homeostatic microglia and APOE-TREM2 was the master pathway in controlling the switch from a homeostatic to a neurodegenerative dysfunctional state of microglia during phagocytosis of apoptotic neurons. This suggested the C1QC+ macrophages exhibited a neurodegenerative dysfunctional phenotype in achalasia, possibly mediating the loss of immune cell homeostasis and neuronal damage.

We took the reviewer's advice and performed in-vitro co-culture experiments of C1QC+ macrophages and neurons. However, owing to the extremely low volume of C1QC+ macrophages in achalasia and especially controls, it is very difficult to dissociate and culture the cells. We tried multiple times to dissociate and culture the C1QC+ macrophages, but all the experiments turned to failed owing to the low amount, contamination, and low cell viability. We admitted this is a main drawback of our study, which we discussed in the Discussion section as follows.

That is "...we mainly found the transcriptional similarity between C1QC+ macrophages and microglia, and showed transcriptional signatures of C1QC+ macrophages in achalasia were similar to those of disease-associated microglia from other neurodegenerative diseases, indicating C1QC+ macrophages exhibit a neurodegenerative dysfunctional phenotype in achalasia. However, no functional experiment was performed to verify the functional similarity between C1QC+ macrophages and microglia in our study. And no functional experiment was explored for the impact of C1QC+ macrophages on myenteric

neurons. Additional investigation with functional study for C1QC+ macrophages and Trms is needed to further explore the mechanism of the pathogenesis of achalasia in the future.”

8) Fig. 4g – How exactly were the Trms defined by the authors? Why were the markers CD4, CD8, CD69 and CD103 chosen for IHC staining? Are the Trm cells really mainly restricted to the myenteric plexus area? It is not obvious from the pictures presented and should be quantified by measuring, for example, the distance between Trms cells and PGP9.5+ cells. How did the Trms localization look like in control patient group? Is the myenteric plexus localization of Trms specific for the achalasia patients only?

Response: Thanks. For the Trms, we validated the Trms by positive expressions of CD3, CD4, and CD69, or CD3, CD8, and CD69 in multicolor IHC. CD69 is the most accepted canonical marker to distinguish the Trms^{11,12}. Besides, we also found a significant overlap of our gene sets with published signatures of Trms in mouse and human (Figure S5a), which could also validate the Trms definition in our study. We validated the higher expression of Trms in the LES of achalasia compared with controls by histological quantifications of multicolor IHC (Figures 4g and 4h). The highly expressed Trms were mainly located surrounding the myenteric plexus of the LES in achalasia.

In our previous submission, besides CD69, we also added CD103 in the panel, which was also reported as a canonical marker of Trms. However, although CD103 is important for the function of Trms, it was not expressed in all Trms both in our and previous scRNA-seq studies (Figure S5c)¹¹⁻¹³. Therefore, we deleted CD103 and only used CD69 to distinguish the Trms in our revised submission.

It is widely reported that loss of myenteric neurons and immune cell infiltration in the LES is the main pathophysiology in achalasia^{26,27}. In achalasia, these myenteric neurons are decreased in number or even absent. Mostly only residual sparse neurons or nerve fibers could be seen in the LES in achalasia.

Although its cause remains largely unknown, ganglionitis resulting from an aberrant immune response has been proposed to underlie the loss of LES myenteric neurons^{26,27}. Therefore, we took images of residual sparse neurons and nerve fibers (shown by PGP9.5) in our figure, which is easily seen in achalasia. In our previous study, we also found CD8+ T cells infiltrated around residual ganglia in the LES in achalasia³¹, in accordance with our current study.

Since the myenteric neurons and ganglia are decreased in number or usually almost absent in achalasia^{26,27}, it is difficult to measure the distance between Trms and myenteric neurons and ganglia. But even so, we could also see Trms were more localized surrounding the residual sparse neurons and nerve fibers (shown by PGP9.5) in achalasia than controls. Since the LES tissue was normal without inflammation in controls, Trms existed, by different with those in achalasia, they were less and equally localized in the LES tissue and not specifically aggregated around the myenteric plexus.

9) More details on the pathophysiology of the disease with illustrations at which part of the esophagus achalasia mainly occurs would help to understand parts of the results and the discussion as it currently is written.

Response: We added an illustration and related descriptions in the Discussion of our study, which included the possible pathophysiology and the target area in the esophagus of achalasia (shown below). Achalasia is a rare motility disorder of the esophagus caused by the gradual degeneration of myenteric neurons. Immune-mediated ganglionitis has been proposed to underlie the loss of myenteric neurons. By scRNA-seq, we found C1QC⁺ macrophages and Trms were significantly expanded and localized surrounding the myenteric plexus in the LES tissue of achalasia. C1QC⁺ macrophages were transcriptionally similar to microglia of the central nervous system and exhibited a neurodegenerative dysfunctional phenotype in achalasia. Trms also presented dysregulated immune responses in achalasia. By potential interactions, the C1QC⁺ macrophages together with Trms might cause the ganglionitis and loss of myenteric neurons. The loss of myenteric neurons could lead to the aberrant esophageal peristalsis, dilated esophagus, and impaired relaxation of the LES, and finally the occurrence of achalasia.

Editorial note: This figure was created by BioRender.com

Minor Points:

1. The researchers identified one cluster of stem cells (Fig. 1b, c), however this a very general term. What type of stem cells was identified?

Response: Thanks for the reviewer's reminder. The CD34+ cells were hematopoietic stem and progenitor cells (HSPCs). We changed the cluster name in the figure and manuscript.

2. Page 8 - "Detailed cluster lineages in different tissue contexts are shown in Figure S1a." - I think the authors wanted to refer to Figure 2b.

Response: Thanks. Since this sentence is wordy and unnecessary, we deleted it during the revision.

3. Page 9 - "We found the functional homogeneity but more heterogeneity among myeloid cells in achalasia". - This reviewer is not sure what the authors intentions are here.

Response: Sorry for the unclear description. We deleted this sentence in the revised manuscript.

4. Fig. 3a -Marta Olah and David Gosselin refers to different microglia gene sets, this is not well described in the Figure caption.

Response: We revised the figure caption as "Comparison of marker genes of myeloid clusters with previously identified human microglial gene sets reported by Marta Olah and David Gosselin."

5. Page 20 - "Previous studies indicated that type I might progress from type II and represent the end stage of achalasia41-43." - what are the characteristics of the end stage of this disease?

Response: Type I might progress from type II and represent the late stage of achalasia. End stage of Type I might be the end stage of this disease. Myenteric neurons have marked depleted or completely disappeared in the LES tissue in patients with end stage disease. As a consequence, no postdeglutitive contractility occurs in the esophagus, and the clinical manifestation is megaesophagus⁴⁴.

We added the description of the end stage of achalasia in the Discussion section.

6. In general, the manuscript requires additional proof reading of a native speaker.

Response: Thanks. The language was double-checked and polished by a native speaker.

Reviewer #3 (Remarks to the Author):

There is still not much known about mechanisms of development of achalasia, an esophageal motility disorder of unknown etiology with pathogenesis involving T-lymphocyte-mediated loss of ganglion cell in the myenteric plexus of the lower esophageal sphincter (LES). Despite the fact that T-lymphocytes, including CD8+ T-cells, have been shown to be involved in inflammation, the precise nature of inflammatory immune cells and their transcriptional profiles are not known.

In this paper the authors describe specific immune cells with unique transcriptional phenotypes using single cell RNA sequencing. In particular they show unique infiltration around the ganglia by C1QC+ macrophages and by specific CD4+ and CD8+ tissue-resident memory T (Trm) cell clusters, such as ZNF683+ CD8+Trm and XCL1+ CD4+Trm. C1QC+ macrophages had high expression of brain microglial gene set and exhibited similar transcripts that are seen in dysfunctional microglia in Alzheimer disease and multiple sclerosis raising a possibility of a role in achalasia pathogenesis. The Trm cell clusters from patients with achalasia, particularly ZNF683+ CD8+ Trm and XCL1+ CD4+Trm, expressed transcripts of activation and proliferation that are typical of inflammatory phenotypes. In addition, these Trm clusters were clonally expanded. These findings are novel and allow a much more nuanced insight into the immune cell type and function in achalasia. The study will be of significance to the field providing essential data needed for furthering the investigation of the pathogenesis of achalasia.

Comments:

1. One general comment is about the use of the term “neurodegenerative disease” or “neurodegenerative” in relation to achalasia. It is not common in medical literature to define achalasia as a neurodegenerative disease. Ganglion loss is thought to be due to inflammation, which is believed by many to be autoimmune. There is no data that I know of describing an intrinsic abnormality of ganglion cells in achalasia to deserve the term ‘neurodegenerative’. I think, it is better to adhere to the terminology which implies that a neurodegenerative disease is an intrinsic disease of neurons leading to their demise.

Response: Thanks. Esophageal achalasia is a rare motility disorder that presents with symptoms including dysphagia, regurgitation, chest pain, and weight loss. Esophageal peristalsis and relaxation of the LES are mediated and coordinated by myenteric neurons. Loss of myenteric neurons and immune cell infiltration in LES is the main pathophysiology in achalasia. Although its cause remains largely unknown, ganglionitis resulting from an aberrant immune response, particularly in genetically susceptible individuals, has been proposed to underlie the loss of LES myenteric neurons^{26,27}.

Among the genetically susceptible factors of achalasia, many neurological candidate genes were reported to mediate the enteric neuron development, structure, and function within the LES, such as nNOS and VIP, GDNF, RET, and SPRY2²⁹. Mutation of these neurological candidate genes were reported to be associated with the development of achalasia, which indicated the enteric neurons were intrinsically disordered independent of immune response in these conditions in achalasia²⁹. Besides, in our study, multiple neurodegenerative and behavioral disorders pathways were associated with C1QC+ macrophages and cDC2s in achalasia, including pathways of neurodegeneration-multiple diseases, AD, PD, and others. We validated the similarity between C1QC+ macrophages and microglia, and found

signatures of DEGs for C1QC+ macrophages were similar to those of disease-associated microglia from other neurodegenerative diseases³⁷⁻⁴³, indicating C1QC+ macrophages exhibit a neurodegenerative dysfunctional phenotype in achalasia. All these findings indicated achalasia could be considered an enteric neurodegenerative disease.

2. Another general comment is about conclusions that suggest a possible “important” or “crucial” role of macrophages or Trm clusters in achalasia. Without data on interactions between the immune cells and ganglion cells it is better to tone down some conclusions to something like: ‘may have a role’ or ‘may be involved’.

Response: Thanks for the reviewer’s advice. We took the reviewer’s advice and changed the related descriptions as “...might be involved in...”.

3. Page 8, last paragraph: “Patients with benign leiomyomas originating from LES served as controls to ensure no invasion of the normal tissue”.

This is unclear. Do the authors mean that they used only normal LES tissue from the specimens with leiomyoma? If so, how far was normal tissue from the leiomyoma? This is important for assessment of the validity of the control tissue.

Response: Thanks. Esophageal leiomyomas are benign tumors of smooth muscle with biologically localized growth and no invasion of surrounding normal tissues. To reduce the impact of leiomyomas at maximum, we took the surrounding normal tissue 5 mm away from the leiomyoma. Therefore, the surrounding normal tissues could consider normal tissues without being influenced by the leiomyomas.

In our recent publication, we compared the achalasia and controls and found CD8+ T cells infiltrated around ganglion in the LES in achalasia³¹. In this study, normal LES tissues were also obtained from patients with esophageal leiomyoma who underwent submucosal tunneling endoscopic resection. We found the CD8+ T cells were very low infiltrated and the ganglion was normal in these control specimens.

We also explained the control tissues more precisely in the manuscript.

4. Page 11. End of first paragraph: “... we speculate that the C1QC+ macrophage cluster was functionally similar to microglia and might have a crucial impact on the ENS.”

Since enteric nervous system was just only once mentioned earlier in the paragraph, I do not think that that there is enough data to speculate that the C1QC+ macrophage cluster “might have a crucial impact on the ENS”.

Response: We changed the sentence as “we speculate that the C1QC+ macrophage cluster was transcriptionally similar to microglia and might have an impact on the ENS.”

5. Page 17. The start of the second paragraph. “Since the clonal expansion of leukocytes is evidence of pathogenesis...”

I think this phrase sounds too broad and vague and is better not used.

Response: Thanks. We took the reviewer’s advice and deleted the sentence.

6. Page 22. The end of the first paragraph: “Since this is the first study to show the importance of LES-infiltrated macrophages in the etiology of achalasia, ...”

It is more likely “pathogenesis” than “etiology”.

Response: Thanks. We took the reviewer’s advice and changed the word.

References

- 1 Viola, M. F. & Boeckxstaens, G. Niche-specific functional heterogeneity of intestinal resident macrophages. *Gut* **70**, 1383-1395, doi:10.1136/gutjnl-2020-323121 (2021).
- 2 Shaw, T. N. *et al.* Tissue-resident macrophages in the intestine are long lived and defined by Tim-4 and CD4 expression. *J Exp Med* **215**, 1507-1518, doi:10.1084/jem.20180019 (2018).
- 3 Bujko, A. *et al.* Transcriptional and functional profiling defines human small intestinal macrophage subsets. *J Exp Med* **215**, 441-458, doi:10.1084/jem.20170057 (2018).
- 4 De Schepper, S. *et al.* Self-Maintaining Gut Macrophages Are Essential for Intestinal Homeostasis. *Cell* **175**, 400-415 e413, doi:10.1016/j.cell.2018.07.048 (2018).
- 5 Domanska, D. *et al.* Single-cell transcriptomic analysis of human colonic macrophages reveals niche-specific subsets. *J Exp Med* **219**, doi:10.1084/jem.20211846 (2022).
- 6 Li, Q. & Barres, B. A. Microglia and macrophages in brain homeostasis and disease. *Nat Rev Immunol* **18**, 225-242, doi:10.1038/nri.2017.125 (2018).
- 7 Bennett, M. L. *et al.* New tools for studying microglia in the mouse and human CNS. *Proc Natl Acad Sci U S A* **113**, E1738-1746, doi:10.1073/pnas.1525528113 (2016).
- 8 Butovsky, O. *et al.* Identification of a unique TGF-beta-dependent molecular and functional signature in microglia. *Nat Neurosci* **17**, 131-143, doi:10.1038/nn.3599 (2014).
- 9 Mildner, A., Huang, H., Radke, J., Stenzel, W. & Priller, J. P2Y(12) receptor is expressed on human microglia under physiological conditions throughout development and is sensitive to neuroinflammatory diseases. *Glia* **65**, 375-387, doi:10.1002/glia.23097 (2017).
- 10 Hickman, S., Izzy, S., Sen, P., Morsett, L. & El Khoury, J. Microglia in neurodegeneration. *Nat Neurosci* **21**, 1359-1369, doi:10.1038/s41593-018-0242-x (2018).
- 11 Szabo, P. A., Miron, M. & Farber, D. L. Location, location, location: Tissue resident memory T cells in mice and humans. *Sci Immunol* **4**, doi:10.1126/sciimmunol.aas9673 (2019).
- 12 Mueller, S. N. & Mackay, L. K. Tissue-resident memory T cells: local specialists in immune defence. *Nat Rev Immunol* **16**, 79-89, doi:10.1038/nri.2015.3 (2016).
- 13 FitzPatrick, M. E. B. *et al.* Human intestinal tissue-resident memory T cells comprise transcriptionally and functionally distinct subsets. *Cell Rep* **34**, 108661, doi:10.1016/j.celrep.2020.108661 (2021).
- 14 Zheng, L. *et al.* Pan-cancer single-cell landscape of tumor-infiltrating T cells. *Science* **374**, abe6474, doi:10.1126/science.abe6474 (2021).
- 15 Mei, Y. *et al.* Single-cell analyses reveal suppressive tumor microenvironment of human colorectal cancer. *Clin Transl Med* **11**, e422, doi:10.1002/ctm2.422 (2021).
- 16 Li, X. *et al.* Single-cell transcriptome profiling reveals the key role of ZNF683 in natural killer cell exhaustion in multiple myeloma. *Clin Transl Med* **12**, e1065, doi:10.1002/ctm2.1065 (2022).
- 17 Lu, Y. C. *et al.* Single-Cell Transcriptome Analysis Reveals Gene Signatures Associated with T-cell Persistence Following Adoptive Cell Therapy. *Cancer Immunol Res* **7**, 1824-1836, doi:10.1158/2326-6066.CIR-19-0299 (2019).
- 18 de Andrade, L. F. *et al.* Discovery of specialized NK cell populations infiltrating human melanoma metastases. *JCI Insight* **4**, doi:10.1172/jci.insight.133103 (2019).
- 19 Fuchs, Y. F. *et al.* Gene Expression-Based Identification of Antigen-Responsive CD8(+) T Cells on a Single-Cell Level. *Front Immunol* **10**, 2568, doi:10.3389/fimmu.2019.02568 (2019).
- 20 Pritchett, J. C. *et al.* High-dimensional and single-cell transcriptome analysis of the tumor microenvironment in angioimmunoblastic T cell lymphoma (AITL). *Leukemia* **36**, 165-176, doi:10.1038/s41375-021-01321-2 (2022).

- 21 Li, J., Zhang, Y., Yang, C. & Rong, R. Discrepant mRNA and Protein Expression in Immune Cells. *Curr Genomics* **21**, 560-563, doi:10.2174/1389202921999200716103758 (2020).
- 22 Liu, Y., Beyer, A. & Aebersold, R. On the Dependency of Cellular Protein Levels on mRNA Abundance. *Cell* **165**, 535-550, doi:10.1016/j.cell.2016.03.014 (2016).
- 23 Greenbaum, D., Colangelo, C., Williams, K. & Gerstein, M. Comparing protein abundance and mRNA expression levels on a genomic scale. *Genome Biol* **4**, 117, doi:10.1186/gb-2003-4-9-117 (2003).
- 24 Rosati, E. *et al.* Overview of methodologies for T-cell receptor repertoire analysis. *BMC Biotechnol* **17**, 61, doi:10.1186/s12896-017-0379-9 (2017).
- 25 Facco, M. *et al.* T cells in the myenteric plexus of achalasia patients show a skewed TCR repertoire and react to HSV-1 antigens. *Am J Gastroenterol* **103**, 1598-1609, doi:10.1111/j.1572-0241.2008.01956.x (2008).
- 26 Boeckxstaens, G. E., Zaninotto, G. & Richter, J. E. Achalasia. *Lancet* **383**, 83-93, doi:10.1016/S0140-6736(13)60651-0 (2014).
- 27 Savarino, E. *et al.* Achalasia. *Nat Rev Dis Primers* **8**, 28, doi:10.1038/s41572-022-00356-8 (2022).
- 28 Azizi, E. *et al.* Single-Cell Map of Diverse Immune Phenotypes in the Breast Tumor Microenvironment. *Cell* **174**, 1293-1308 e1236, doi:10.1016/j.cell.2018.05.060 (2018).
- 29 Gockel, H. R. *et al.* Achalasia: will genetic studies provide insights? *Hum Genet* **128**, 353-364, doi:10.1007/s00439-010-0874-8 (2010).
- 30 Li, Q. *et al.* Whole-exome sequencing reveals common and rare variants in immunologic and neurological genes implicated in achalasia. *Am J Hum Genet* **108**, 1478-1487, doi:10.1016/j.ajhg.2021.06.004 (2021).
- 31 Ma, L. Y. *et al.* A cross-sectional study reveals a chronic low-grade inflammation in achalasia. *J Gastroenterol Hepatol*, doi:10.1111/jgh.16091 (2022).
- 32 Veglia, F., Sanseviero, E. & Gabrilovich, D. I. Myeloid-derived suppressor cells in the era of increasing myeloid cell diversity. *Nat Rev Immunol* **21**, 485-498, doi:10.1038/s41577-020-00490-y (2021).
- 33 Zhang, Q. *et al.* Landscape and Dynamics of Single Immune Cells in Hepatocellular Carcinoma. *Cell* **179**, 829-845 e820, doi:10.1016/j.cell.2019.10.003 (2019).
- 34 Zhao, F. *et al.* S100A9 a new marker for monocytic human myeloid-derived suppressor cells. *Immunology* **136**, 176-183, doi:10.1111/j.1365-2567.2012.03566.x (2012).
- 35 Hong, S. *et al.* Complement and microglia mediate early synapse loss in Alzheimer mouse models. *Science* **352**, 712-716, doi:10.1126/science.aad8373 (2016).
- 36 Bartels, T., De Schepper, S. & Hong, S. Microglia modulate neurodegeneration in Alzheimer's and Parkinson's diseases. *Science* **370**, 66-69, doi:10.1126/science.abb8587 (2020).
- 37 Mathys, H. *et al.* Temporal Tracking of Microglia Activation in Neurodegeneration at Single-Cell Resolution. *Cell Rep* **21**, 366-380, doi:10.1016/j.celrep.2017.09.039 (2017).
- 38 Masuda, T. *et al.* Spatial and temporal heterogeneity of mouse and human microglia at single-cell resolution. *Nature* **566**, 388-392, doi:10.1038/s41586-019-0924-x (2019).
- 39 Mathys, H. *et al.* Single-cell transcriptomic analysis of Alzheimer's disease. *Nature* **570**, 332-337, doi:10.1038/s41586-019-1195-2 (2019).
- 40 Absinta, M. *et al.* A lymphocyte-microglia-astrocyte axis in chronic active multiple sclerosis. *Nature* **597**, 709-714, doi:10.1038/s41586-021-03892-7 (2021).
- 41 Keren-Shaul, H. *et al.* A Unique Microglia Type Associated with Restricting Development of Alzheimer's Disease. *Cell* **169**, 1276-1290 e1217, doi:10.1016/j.cell.2017.05.018 (2017).
- 42 Olah, M. *et al.* A transcriptomic atlas of aged human microglia. *Nat Commun* **9**, 539, doi:10.1038/s41467-018-02926-5 (2018).

- 43 Krasemann, S. *et al.* The TREM2-APOE Pathway Drives the Transcriptional Phenotype of Dysfunctional Microglia in Neurodegenerative Diseases. *Immunity* **47**, 566-581 e569, doi:10.1016/j.immuni.2017.08.008 (2017).
- 44 Kahrilas, P. J. & Boeckxstaens, G. The spectrum of achalasia: lessons from studies of pathophysiology and high-resolution manometry. *Gastroenterology* **145**, 954-965, doi:10.1053/j.gastro.2013.08.038 (2013).

REVIEWERS' COMMENTS

Reviewer #1 (Remarks to the Author):

I very much appreciate the thoroughness of the authors to reviewer's critics. All of my concerns were addressed.

Reviewer #2 (Remarks to the Author):

Dear authors,

I'm impressed by the amount of work and level of detail that went into the revised version of your manuscript.

I went through the corrections you made in response to my comments and agree with the new storyline and experimental line-up.

With this I consider the manuscript sufficiently improved to meet the criteria of publication in Nature Communications.

Reviewer #3 (Remarks to the Author):

I consider the response to item 1 unsatisfactory.

Rare cases of familial achalasia with gene mutations including genes with neurological function do not make achalasia a neurodegenerative disease. In addition, similarity of features between C1QC+ macrophages those of microglia from neurodegenerative diseases found in this manuscript also does not make achalasia a neurodegenerative disease. Therefore, achalasia cannot be considered an enteric neurodegenerative disease based on the these data, as suggested by the authors.

The response to items 2-6 is satisfactory

Reviewer #1 (Remarks to the Author):

I very much appreciate the thoroughness of the authors to reviewer's critics. All of my concerns were addressed.

Response: Thanks for the reviewer's comments.

Reviewer #2 (Remarks to the Author):

Dear authors,

I'm impressed by the amount of work and level of detail that went into the revised version of your manuscript. I went through the corrections you made in response to my comments and agree with the new storyline and experimental line-up. With this I consider the manuscript sufficiently improved to meet the criteria of publication in Nature Communications.

Response: Thanks for the reviewer's comments.

Reviewer #3 (Remarks to the Author):

1. I consider the response to item 1 unsatisfactory.

Rare cases of familial achalasia with gene mutations including genes with neurological function do not make achalasia a neurodegenerative disease. In addition, similarity of features between CIQC+ macrophages those of microglia from neurodegenerative diseases found in this manuscript also does not make achalasia a neurodegenerative disease. Therefore, achalasia cannot be considered an enteric neurodegenerative disease based on these data, as suggested by the authors.

Response: Thanks for the reviewer's comments. Loss of myenteric neurons and immune cell infiltration in LES is the main pathophysiology in achalasia. Ganglionitis resulting from an aberrant immune response, particularly in genetically susceptible individuals, has been proposed to underlie the loss of LES myenteric neurons^{1,2}. Although with some similar features between achalasia and neurodegenerative diseases, we agree with the reviewer's advice that achalasia cannot be considered an enteric neurodegenerative disease based on the current data. Therefore, in the background, we changed the sentence "Since achalasia is an immune-mediated neurodegenerative disease....." to "Since the loss of myenteric neurons and ganglionitis was the main pathophysiology in achalasia....."

2. The response to items 2-6 is satisfactory.

Response: Thanks.

References

1. Boeckstaens, G.E., Zaninotto, G. & Richter, J.E. Achalasia. *Lancet* **383**, 83-93 (2014).
2. Savarino, E., *et al.* Achalasia. *Nat Rev Dis Primers* **8**, 28 (2022).